# Learning to cluster neuronal function

**Nina S. Nellen,**[1,†,*] **Polina Turishcheva,**[1,†,*] **Michaela Vystrčilová,**[1]
**Shashwat Sridhar,**[2,3] **Tim Gollisch,**[2–5] **Andreas S. Tolias,**[6–9] **Alexander S. Ecker**[1,10,*]

[1] Institute of Computer Science and Campus Institute Data Science, University Göttingen, Germany
[2] University Medical Center Göttingen, Department of Ophthalmology, Göttingen, Germany
[3] Bernstein Center for Computational Neuroscience Göttingen, Göttingen, Germany
[4] Cluster of Excellence "Multiscale Bioimaging: from Molecular Machines to Networks of Excitable Cells" (MBExC), University of Göttingen, Germany
[5] Else Kröner Fresenius Center for Optogenetic Therapies, University Medical Center Göttingen, Germany
[6] Department of Ophthalmology, Byers Eye Institute, Stanford University School of Medicine, CA, US
[7] Stanford Bio-X, Stanford University, Stanford, CA, US
[8] Wu Tsai Neurosciences Institute, Stanford University, CA, US
[9] Department of Electrical Engineering, Stanford University, CA, US
[10] Max Planck Institute for Dynamics and Self-Organization, Göttingen, Germany

[†] Shared contribution
[*]`nina.nellen@uni-goettingen.de`
[*]`{turishcheva,ecker}@cs.uni-goettingen.de`

## Abstract

Deep neural networks trained to predict neural activity from visual input and behaviour have shown great potential to serve as digital twins of the visual cortex. Per-neuron embeddings derived from these models could potentially be used to map the functional landscape or identify cell types. However, state-of-the-art predictive models of mouse V1 do not generate functional embeddings that exhibit clear clustering patterns which would correspond to cell types. This raises the question whether the lack of clustered structure is due to limitations of current models or a true feature of the functional organization of mouse V1. In this work, we introduce DECEMber – Deep Embedding Clustering via Expectation Maximization-based refinement – an explicit inductive bias into predictive models that enhances clustering by adding an auxiliary $t$-distribution-inspired loss function that enforces structured organization among per-neuron embeddings. We jointly optimize both neuronal feature embeddings and clustering parameters, updating cluster centers and scale matrices using the EM-algorithm. We demonstrate that these modifications improve cluster consistency while preserving high predictive performance and surpassing standard clustering methods in terms of stability. Moreover, DECEMber generalizes well across species (mice, primates) and visual areas (retina, V1, V4). The code is available at `https://github.com/Nisone2000/DECEMber`, `https://github.com/ecker-lab/cnn-training`.

## 1 Introduction

Understanding whether neurons form discrete cell types or lie on a continuum is a fundamental question in neuroscience [1]. Previous research has extensively investigated the morphological and electrophysiological properties of neurons in the visual cortex. While discrete anatomical and transcriptomic classifications have been proposed [2–4], recent work on the mouse brain suggests a

39th Conference on Neural Information Processing Systems (NeurIPS 2025).

more continuous organization [5, 6]. Significantly less attention has been devoted to the neurons' functional properties. Each neuron can be characterized by a function that maps high-dimensional sensory inputs to its one-dimensional neuronal response. These functions are highly non-linear, making their analysis complex. Discrete functional cell types are well established in the retina [7] but their existence remains unclear in the mouse visual cortex.

Recently, deep networks showed great potential for predicting neural activity from sensory input [8–15] and also in inferring novel functional properties [16–18]. These networks learn per-neuron vectors of parameters, which are interpreted as neuronal functional embeddings. There were several attempts to use these embeddings to reveal the underlying structure of neuronal population functions through unsupervised clustering [13, 19–21]. However, in none of these studies well-separated clusters emerged, raising the question of whether distinct functional cell types exist among excitatory neurons in the mouse visual cortex. A central challenge is cluster consistency: How reliably are neurons grouped into the same cluster across different model runs? Clustering metrics such as the Adjusted Rand Index (ARI) [22], which evaluates cluster assignment agreement across different seeds or clustering methods and similarity metrics between individual neurons' remained relatively low [13]. These low scores show that clustering results lack the stability and distinctiveness necessary to making strong claims about biological interpretations.

In this work, we incorporate an explicit clustering bias into the training of neuronal embeddings to improve the identifiability of functional cell types, One could view it as model-driven hypothesis testing: if clear functional cell types exist then such bias should improve the model performance, embeddings structure and/or cluster consistencies.

To improve the cluster separability of neuronal embeddings we took inspiration from Deep Embedding Clustering (DEC) [23] and introduced a new clustering loss, which combines updating clusters' locations and shapes along with learning feature representations. We measured the consistency of clustered features across models fitted on different seeds by computing ARI on their clustering results. Additionally, we examined how the clustering loss strength influenced models' performance.

Our contributions are

- We adapted the DEC-loss [23] to allow for non-isotropic multivariate clusters of different sizes by learning a multivariate $t$ mixture model [24].
- We improved cluster consistency while maintaining a state-of-the-art predictive model performance.
- We showed that our method generalizes well, improving cluster consistency across different species, visual areas, and model architectures.

## 2   Background and related work

**Predictive models for visual cortex.**   In comparison to task-driven networks [25–28], pioneering data-driven population models [10, 29, 30] introduced the core-readout framework, which separates the stimulus-response functions of neurons into a shared nonlinear feature space (core) and per-neuron specific set of linear weights – the readout. The core is shared among all neurons and outputs a nonlinear set of basis functions spanning the feature space of the neuronal nonlinear input-output functions of dimension (height $\times$ width $\times$ feature channels). The early models were extended by including behavioral modulation [17, 31], latent brain state [11, 32] or the perspective transformations of the eye [12]. The core architecture was improved by introducing biological biases such as a rotation-equivariant core [14] to account for orientation selectivity in V1 neurons [33], extending to dynamic models with video input [9, 12, 15, 18, 31] or using transformer architectures [34].

Klindt et al. [35] introduced a factorized readout for each neuron, comprising a spatial mask $M_n$ specifying its receptive field (RF) position and feature weights. This approach was refined by Lurz et al. [36], who proposed the Gaussian readout, replacing the full spatial mask with a pair of coordinates $(x_n, y_n)$ drawn from a learned normal distribution. To predict the neuronal response the model computes the dot product between the neuron's weight vector (per-neuron embedding) and each feature map at the RF location. For later visual layers, like V4, the receptive field location is not necessarily fixed. Therefore, Pierzchlewicz et al. [37] introduced an attention readout, which indicates the most important feature locations for a neuron $n$ depending on the input image.

While different readouts exist, few works have examined their consistency. Turishcheva et al. [13] showed that factorized readouts produced more consistent neuronal clusters than Gaussian readouts,

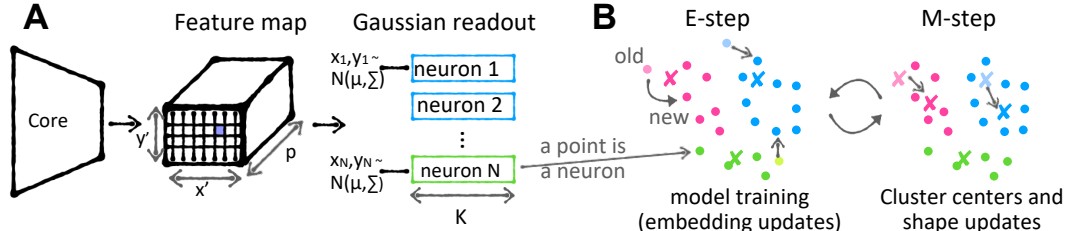

Figure 1: **A: Model architecture:** The model consists of a neuronwise shared core outputting a feature map of size (height × width × feature channels) and neuron specific Gaussian readouts. They consist of a receptive field position and a weight vector. The RF position chooses the vector in the feature map which is then combined with the neuron's weight vector by a dot product to get the neuron's response. **B: Clustering procedure:** We're clustering the readouts with an additional loss to incorporate the cluster bias into the features. We update the clustering parameters (cluster centers and scale matrices) with an EM step of a $t$ mixture model as in Alg. 1.

despite lower predictive performance. They addressed this by introducing adaptive log-norm regularization to balance model expressiveness and feature consistency. However, the ARI scores were still not high enough to claim distinct cell types. Moreover, their work involved a rotation-equivariant convolutional core and required a post-hoc alignment procedure [38] to interpret the cluster structures.

**Deep embedding clustering.** Deep Embedding Clustering (DEC) [23] combines clustering with representation learning. It introduced a clustering loss that simultaneously drives learning the cluster centroids and encourages the feature representation to separate the clusters. After pretraining a deep autoencoder without the clustering loss, the cluster centers $\mu_j$ are initialized using k-means [39]. DEC then minimizes a Kullback-Leibler (KL) divergence of soft cluster assignments $Q$ and an auxiliary target distribution $P$ defined as follows:

$$q_{ij} = \frac{\left(1 + \frac{\|z_i - \mu_j\|^2}{\nu}\right)^{-\frac{\nu+1}{2}}}{\sum_{j'} \left(1 + \frac{\|z_i - \mu_{j'}\|^2}{\nu}\right)^{-\frac{\nu+1}{2}}} \quad (2.1) \qquad p_{ij} = \frac{q_{ij}^2 / f_j}{\sum_{j'} q_{ij'}^2 / f_{j'}} \quad \text{with } f_j = \sum_i q_{ij}. \quad (2.2)$$

The $q_{ij}$s are the probabilities of sample $z_i$ belonging to cluster $j$ and are represented by a Student's $t$-distribution with unit scale and degree of freedom $\nu$ being set to 1. The target distribution $P$ (Eq. (2.2)) is chosen such that it:

- **"strengthens predictions."** Original values $q_{ij}$ denote the soft assignment probability of a data point $i$ belonging to cluster $j$. Squaring $q_{ij}$ and then re-normalizing makes high-confidence assignments more dominant while further diminishing the influence of low-confidence ones.
- **"emphasizes high-confidence data points."** A high $q_{ij}$ dominates $q_{ij}^2 / f_j$, meaning that points strongly associated with a cluster contribute more to $p_{ij}$.
- **"normalizes loss contribution of each centroid to prevent large clusters from distorting the hidden feature space."** Without $f_j$, larger clusters could dominate the feature space since they would contribute disproportionately to the loss. By dividing by $f_j$ the impact of each cluster is normalized, ensuring that smaller clusters are not overshadowed by larger ones.

Guo et al. [40] extended this approach by jointly optimizing the clustering objective and the autoencoder's reconstruction loss, enabling the model to learn clusters while preserving the local structure of the feature space.

## 3 DECEMber – Deep Embedding Clustering via Expectation Maximization-based refinement

DECEMber combines training a predictive model of neuronal responses with the learning of a clustered feature embeddings by optimizing a loss inspired by Deep Embedding Clustering and iteratively updating cluster parameters using the EM algorithm. We now describe our approach (illustrated in Fig. 1, described in Alg. 1).

---

**Algorithm 1** Model Training with clustering loss

---

**Inputs:** Degrees of freedom $\nu$, clustering weight $\beta$, core parameters $\theta$, neuronal embeddings (readout) $Z$

**Output:** Parameters $\mu_j, \Sigma_j, \theta$ and $Z$

**Pretraining:** Train the predictive model by optimizing $L_{\text{model}}$ w.r.t. $\theta$ and $Z$ for $m$ epochs

**Initialize:** Cluster centers $\mu_j$ with $k$-means and diagonal scale matrix $\Sigma_j$ as within-cluster variance

**for** epoch $t = 1$ to $T$ **do**

    **for** minibatch $b$ in dataset **do**

        **(1) E-step (Expectation):** Compute

          1.1 Soft assignments $q_{ij} = \frac{f_t(z_i; \mu_j, \Sigma_j, \nu)}{\sum_{j'=1}^{J} f_t(z_i; \mu_{j'}, \Sigma_{j'}, \nu)}$         (3.2)

          1.2 Latent scales $u_{ij} = \frac{\nu + K}{\nu + (z_i - \mu_j)' \Sigma_j^{-1} (z_i - \mu_j)}$         (3.3)

        **(2) M-step (Maximization):** Update parameters

          2.1 Update $\mu_j = \frac{\sum_{i=1}^{N} q_{ij} u_{ij} z_i}{\sum_{i=1}^{N} q_{ij} u_{ij}}$         (3.4)

          2.2 Update $\Sigma_j = \frac{\sum_{i=1}^{N} q_{ij} u_{ij} (z_i - \mu_j)(z_i - \mu_j)'}{\sum_{i=1}^{N} q_{ij}}$         (3.5)

        **(3) Gradient step:** Optimize predictive model parameters

          3.1 Minimize $L = L_{\text{model}} + \beta KL(Q||P)$ w.r.t $\theta, Z$

          with $p_{ij} = \frac{q_{ij}^2 / f_j}{\sum_k q_{ik} / f_k}$ and $f_j = \sum_i q_{ij}$

    **return** $\mu, \Sigma, \theta, Z$

---

**Predictive model for visual cortex.** We build on a state-of-the-art predictive model [8] for responses $r_i$ of neurons $i = 1, ..., N$ to visual stimuli $s \in \mathbb{R}^{H' \times W' \times T \times C}$. Here $H'$ and $W'$ are height and width of the input, $T$ time if the input is a video and $C$ is the amount of channels: $C = 1$ for grayscale or $C = 3$ for RGB. For static visual input (images), $T = 1$ and could be ignored. If behavior variables – such as pupil size, locomotion speed, and changes in pupil size – are present, they are concatenated to the stimuli as channels [8]. The model combines a shared convolutional core $\Phi$ with neuron-specific Gaussian readouts $\psi_i$ (Fig. 1A). The core outputs a feature space $\Phi(s) \in \mathbb{R}^{H \times W \times K}$. We denote the core's parameters by $\theta$. The readout [36] $\psi_i : \mathbb{R}^{H \times W \times K} \mapsto \mathbb{R}$ first selects the features from $\Phi$ at the neuron's receptive field location $(x_i, y_i)$ using bilinear interpolation, which we write with a slight abuse of notation as $\Phi(\tilde{s})_{x_i y_i} \in \mathbb{R}^K$, resulting in a feature vector $\phi_i \in \mathbb{R}^K$. It then computes the predicted neuronal response $\hat{r}_i = z_i^T \phi_i$, where $z_i \in \mathbb{R}^K$ is the neuron-specific readout weight (its functional embedding), overall

$$\hat{r}_i(s) = \psi_i(\Phi(s)) = z_i^T \Phi(\tilde{s})_{x_i y_i}. \tag{3.1}$$

**EM step to update cluster parameters.** Instead of directly learning the cluster centroids via gradient descent, we updated them after each batch using the EM algorithm applied to the Student's $t$-mixture model, $f_{\text{TMM}}(z_i; \Theta) = \frac{1}{J} \sum_{j=1}^{J} f_t(z_i; \mu_j, \Sigma_j, \nu)$, [24] where degree of freedom $\nu$ controls the probability mass in the tails (if $\nu \to \infty$ the $t$-distribution becomes Gaussian). The density of the multivariate Student's $t$-distribution is:

$$f_t(z_i; \mu_j, \Sigma_j, \nu) = \frac{\Gamma\left(\frac{\nu+K}{2}\right)}{\Gamma\left(\frac{\nu}{2}\right) \nu^{\frac{p}{2}} \pi^{\frac{p}{2}} |\Sigma_j|^{\frac{1}{2}}} \left(1 + \frac{1}{\nu}(z_i - \mu_j)^T \Sigma_j^{-1}(z_i - \mu_j)\right)^{-\frac{\nu+K}{2}} \tag{3.6}$$

$$= \int_0^\infty \mathcal{N}(z_i \mid \mu_j, \tfrac{1}{u}\Sigma_j) \cdot \text{Gamma}\left(u \mid \tfrac{\nu}{2}, \tfrac{\nu}{2}\right) \, du. \tag{3.7}$$

with the latter being the so-called shape-rate form of the $t$-distribution [41]. This interpretation is useful because introducing the Gamma-distributed latent variable $u$ allows closed-form M-step updates for $\mu_j$ and $\Sigma_j$, whereas direct likelihood optimization in a $t$-mixture model does not generally admit closed-form solutions.

The full procedure is summarized in Alg. 1 and alternates between: (1) E-Step: Compute soft cluster assignments $q_{ij}$ (Eq. (3.2)) – the probability of feature $i$ belonging to cluster $j$ – and the latent scaling factors $u_{ij}$ (Eq. (3.3)). (2) M-Step: Update cluster means $\mu_j$ (Eq. (3.4)) and (diagonal) scale matrices

$\Sigma_j$ (Eq. (3.5)). (3) Gradiet step: Update the parameters of the core and readout via one iteration of stochastic gradient descent.

**Clustering loss on readout weights.** To encourage a well-clustered structure on the neuron-specific readout weights, we augment the standard model loss with a clustering objective. Specifically, we minimize the KL divergence between soft cluster assignments $q_{ij}$ (Eq. (3.2)) and target distributions $p_{ij}$ (Eq. (2.2)):

$$L_{\text{cluster}} = \text{KL}(Q(Z) \parallel P(Z)) = \sum_{i=1}^{N} \sum_{j=1}^{J} p_{ij} \log \left( \frac{p_{ij}}{q_{ij}} \right). \tag{3.8}$$

This auxiliary loss encourages the embeddings to form $J$ distinct clusters.

Xie et al. [23] use a pretrained autoencoder with well-separated embeddings and model soft cluster assignments using a Student's $t$-distribution with fixed unit scale. However, this setup is too constrained for our regression setting, where the mean and scale of the embeddings $z_i$ are restricted by the regression loss. By adopting a TMM, where each cluster is characterized by both its center $\mu_j$ and scale matrix $\Sigma_j$, we allow the clustering structure to adapt more flexibly during training.

## 4 Experiments

**Clustering loss hyperparameters.** For the clustering loss, we fixed the degrees of freedom to $\nu = 2.1$, just above the threshold where the variance $\frac{\nu}{\nu-2}\Sigma$ becomes defined (only for $\nu > 2$). To balance model flexibility and robustness, we allowed each cluster to have its own diagonal scale matrix $\Sigma_j$, which alloed for different variances per embedding dimension while preventing overfitting. For each dataset, we adjusted the clustering strength $\beta \in \mathbb{R}$ such that it is in the same order of magnitude as the model loss at initialization.

**Pretraining and cluster initialization.** Before adding the clustering loss, we pretrained the baseline model for $m$ epochs, such that the model could already predict the responses reasonably well. We explored $m = 5, \ldots, 40$ to assess how the length of pretraining (PE) affected our results. We followed Turishcheva et al. [13] for the pretraining procedure, minimizing the following loss:

$$L_{\text{model}} = L_{\text{P}} + L_{\text{reg}} = \frac{1}{N} \sum_{l=1}^{L} \sum_{i=1}^{N} (\hat{r}_{il} - r_{il} \log \hat{r}_{il}) + L_{\text{reg}} \tag{4.1}$$

where $L_{\text{P}}$ is the Poisson loss that aligned per-image $l = 1, \ldots, L$ model predictions $\hat{r}_{il}$ with observed neuronal responses $r_{il}$ since neuron's firing rates follow a Poisson process [42], and $L_{\text{reg}}$ is the adaptive regularizer that was shown to result in improved embedding consistency [13].

After pretraining, we initialized the cluster centroids $\mu_j$ with k-means [39] and the diagonal scale matrices $\Sigma_j$ as the within-cluster variances. We continued training using $L_{model} + \beta L_{\text{cluster}}$, with $L_{cluster}$ as in Eq. (3.8) and scaled with $\beta$.

**Evaluation of model performance.** Building on previous research [8, 14, 16, 34, 43, 44], we evaluated the model's predictive performance by computing the Pearson correlation (across images in the test set) between the measured and predicted neural responses, averaged across neurons.

**Evaluation of embedding consistency.** We wanted to assess the relative structure of the embedding space: Do the same groups of neurons consistently cluster together across models fit with different initial conditions? To quantify this notion, we took DECEMber's cluster assignments and measured how often neuron pairs are assigned to the same group using the Adjusted Rand Index (ARI) [22], which quantifies the similarity between two clustering assignments, $X$ and $Y$. The ARI remains unchanged under permutations of cluster labels. ARI equals one if and only if the two partitions are identical and it equals zero when the partitions agreement is no better than random.

To compare DECEMber with a baseline, we extracted neuronal embeddings from the fully converged default model and fitted Gaussian Mixture Models (GMMs) using the same number of clusters as DECEMber, diagonal covariance, and a regularization of $10^{-6}$. We then computed the ARI across three GMM partitions from baseline models initialized with different seeds, using a fixed GMM seed.

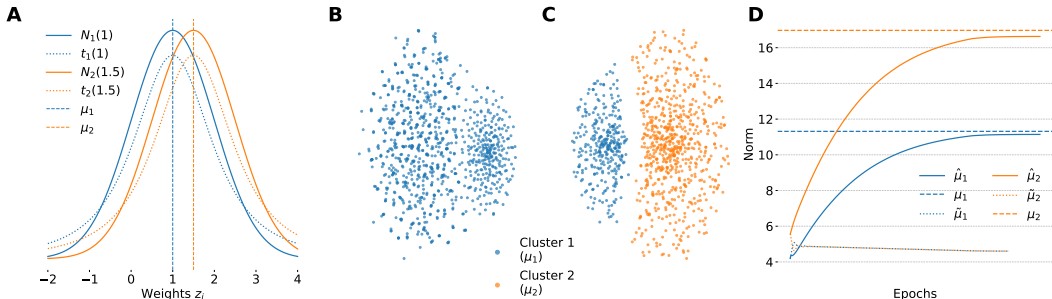

Figure 2: **A:** PDF of $z_1$ (blue) and $z_2$ (orange) of the underlying true normal distribution and $t$-distribution with unit scale estimated by DEC-loss. The two $t$-distributions as well as the normal distributions are highly overlapping. **B+C** $t$-SNE projection of toy data after training with DEC loss (B) vs DECEMber (C). We first pretrain a simple linear regression model by minimizing an MSE-loss for 30 epochs. Then we are training jointly with MSE and KL loss. **B:** Visualization of clustering with the DEC loss. All features are assigned to one cluster. Clustered structure is still visible. **C:** Clustering of the learned features with DECEMber. All features get assigned to the right cluster. **D:** Norms of learned cluster centers $\tilde{\mu}_1$ and $\tilde{\mu}_2$ for the DEC-loss. It is clearly visible that the cluster centers collapse after only a few iterations whereas updated cluster centers via DECEMber $\hat{\mu}_1$ and $\hat{\mu}_2$ are converging towards their true mean $\mu_1$ and $\mu_2$, with $\|\mu_1\|_2 = \sqrt{128} \approx 11.3$ and $\|\mu_2\|_2 = \sqrt{128 \cdot 1.5^2} \approx 17$.

**Visualization.** To visualize the neuronal embeddings, we employed t-SNE [45], following the guidelines of [46]. Specifically, we set the perplexity to $N/100$, the learning rate to 1 and early exaggeration to $N/10$. To be comparable with prior work [13, 20], we randomly sample 2,000 neurons from each of the seven mice in the dataset and used the same neurons across all visualizations.

## 5 Results

**DEC-loss needs learned scale: toy example illustration.** To assess whether the DEC-loss provides a useful clustering when applied to model weights that are restricted by a regression loss instead of autoencoder embeddings, we constructed a simple toy example consisting of linear neurons whose responses are given as $y_{ij} = z_i^T x_j + \epsilon_{ij}$, where $z_i$ are the neurons' weights, $x_j$ the stimuli and $\epsilon_{ij}$ Gaussian noise. We generated 1100 white noise stimuli, each of the form $x_j \in \mathbb{R}^{128}$ with $x_j \sim \mathcal{N}(0, 1)$. We generated 1000 neuronal embeddings $z_i$ such that they would naturally form 2 clusters. We created 300 weights of the form $z_i \sim \mathcal{N}(I_{128}), I_{128})$ and 700 $z_i \sim \mathcal{N}(1.5I_{128}, I_{128})$. To finally get the neuronal responses we sampled Gaussian noise around 0 with a variance that matches a chosen signal to noise ratio (SNR). In the here shown example we used SNR=2. A detailed discussion about SNR can be found in the Appendix A.1.

We pretrained a linear regression model on this data for 30 epochs by minimizing the MSE of predicted and learned responses. After that we continued training, by jointly minimizing the KL divergence on the learned centers and the MSE using the DEC-loss versus DECEMber. We used early stopping as well as a learning rate scheduler. In theory the model should learn the weights of the two clusters centered at $\mu_1 = (1, \ldots, 1) \in \mathbb{R}^{128}$ and $\mu_2 = (1.5, \ldots, 1.5) \in \mathbb{R}^{128}$ and assign 300 neurons to cluster 1 and 700 neurons to cluster 2.

We found that the vanilla DEC loss fails to identify clusters even in this simple toy example, where cluster weights are well-separable after pretraining. This is because DEC employs a Student's-$t$ distribution with a fixed unit scale parameter for all clusters, which is too large given how close the two weight distributions of clusters 1 and 2 are (Fig. 2A). As the magnitude of the weights is given by the regression problem, the scale of the $t$ distribution needs to be adjusted appropriately during training. When this is not done (as in vanilla DEC), the cluster centroids $\tilde{\mu}_1$ and $\tilde{\mu}_2$ rapidly collapse to a single point after only a few iterations (Fig. 2D). Even though in this toy example the true covariance of the clusters has unit scale after the short pretraining phase the within cluster variance is much lower leading to a collapse of the DEC-centers. This happens because there exists a degenerate optimum of the KL divergence: If all cluster centers are equal $\mu_1 = \ldots = \mu_J = c$, plugging them into Eq. (2.1)

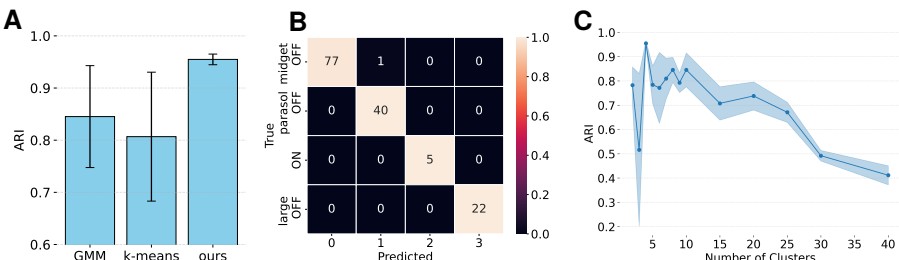

Figure 3: **A**: RI across 3 seeds for GMM, k-means and DECEMber. **B**: DECEMber predictions. Pretraining length: 25 epochs. Corresponding test correlation: $0.805 \pm 0.068$ (std). **C**: ARI across 3 seeds for DECEMber over different number of clusters. ARI shows a clear peak at 4 clusters.

$q_{ij} = \frac{\left(1+\|z_i-c\|^2\nu^{-1}\right)^{-(\nu+1)/2}}{\sum_{j'}(1+\|z_i-c\|^2\nu^{-1})^{-(\nu+1)/2}} = \frac{1}{J}$ gives us $p_{ij} = q_{ij}^2/(\sum_{j'} q_{ij'}^2) = (1/J^2)/(J \cdot (1/J^2)) = 1/J$ as $f_j = f_{j'}$, (Eq. (2.2)) which means the KL divergence $KL(P\|Q) = 0$, which of course is not a meaningful solution. In DEC, this minimum is not usually found in practice because clusters are initialized with sufficient separation after pretraining.

To avoid this collapse, we instead used a multivariate $t$-distribution with (diagonal) scale matrices $\Sigma_j$ for each cluster, updating both position and scale with an EM step (Alg. 1). On the same toy example, this approach succeeded in finding good cluster separation (Fig. 2C), and the cluster centers $\hat{\mu}_1$ and $\hat{\mu}_2$ converged towards the true underlying locations (Fig. 2D).

**DECEMber accurately classifies retinal ganglion cells and outperforms conventional clustering approaches.** To check whether DECEMber works on real data, we applied it on marmoset retinal ganglion cells (RGCs) where the existence of discrete cell types is well established [47]. We used data from two male marmoset retinas published by Sridhar et al. [48], where the neural activity was recorded using a micro-electrode array while presenting grayscale natural movies.

As we observed substantial differences between the two retinas' temporal response features (potentially due to temperature variation [49]), we followed Vystrčilová et al. [15] and trained a separate model for each retina to avoid clustering by retina. We trained the model on all reliably responding cells ($N = 235$). However, not all of them corresponded to a known primate RGC type and thus were not assigned a cell type label. When evaluating DECEMber, we only used the labeled cells. The first retina contained responses of four cell types (78 midget-OFF-like cells, 40 parasol-OFF-like cells, 5 ON-like cells, and 22 large-OFF cells, further details on classification are in App. A.2).

We trained a baseline version of a CNN model [15] separately without our proposed clustering loss, using three random seeds. Baseline clustering was then performed post hoc using GMM and k-means. Subsequently, we continued training the model with our clustering loss, again using three seeds.

Applied to marmoset retina data, DECEMber achieved reliable classification across cell types, with high clustering consistency (ARI = $0.96 \pm 0.01$) for 4 clusters while maintaining a high predictive performance of $0.81 \pm 0.07$. It surpassed both GMM and k-means, (Fig. 3A) effectively separating even highly unbalanced groups, such as the ON-cells, resulting in an almost perfect confusion matrix (Fig. 3B). In contrast to k-means, which is sensitive to initialization (Suppl. Fig. 12), DECEMber exhibited greater robustness and aligned more closely with the ground truth labels while the model retained high predictive accuracy. When we tested DECEMber for different amount of clusters we can see a clear peak in ARI at the true number of clusters 4 (Fig. 3C).

**DECEMber enhances local structure among embeddings and hurts performance once it dominates the overall model loss.** Next, we asked if DECEMber could help to find functional cell types in a visual area without clear known cluster separation. We used SENSORIUM 2022 dataset and baseline architecture to train a model to predict responses of mouse primary visual cortex to grayscale images. for seven mice (more detail on data in App. A.5). Previous work [13, 20] observed density modes in the functional embeddings of mouse V1 neurons (Fig. 4A) and hypothesized that these modes may correspond to discrete functional cell types. To investigate whether these patterns reflect true discrete and distinct cell types, we applied DECEMber (Alg. 1), hypothesizing that if such types exist, DECEMber would help to separate them. As the number of excitatory cell types in the mouse

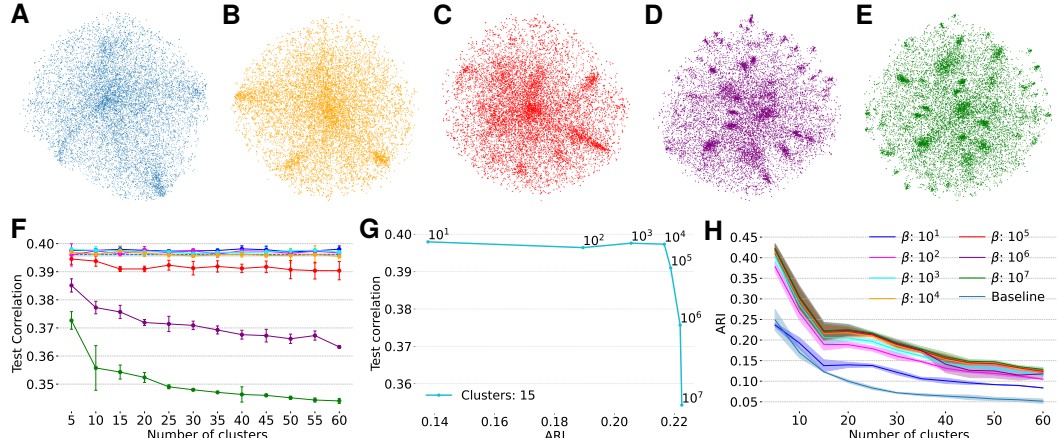

Figure 4: **A**: $t$-SNE of baseline model without clustering loss. **B-E**: $t$-SNE projections of our model with clustering bias for different multipliers $\beta$ and tuned learning rates (lr). All models use 15 clusters, PE = 10 and seed 100. **B**: $\beta = 10^4$ and $lr = 0.008$. **C**: $\beta = 10^5$ and $lr = 0.008$. **D** $\beta = 10^6$ and $lr = 0.007$. **E**: $\beta = 10^7$ and $lr = 0.003$. **F**: Corresponding model performances of the models with clustering bias and differing weights $\beta$, tuned learning rates as described in B-D. **G**: Predictive performance vs. ARI for different $\beta$. We can see that ARI increases with the increase of $\beta$ until the model performance drops. After that ARI doesn't increase further. Here we fixed the number of clusters to 15. **H**: ARI for different clustering weights $\beta$ with optimal learning rates, PE = 10.

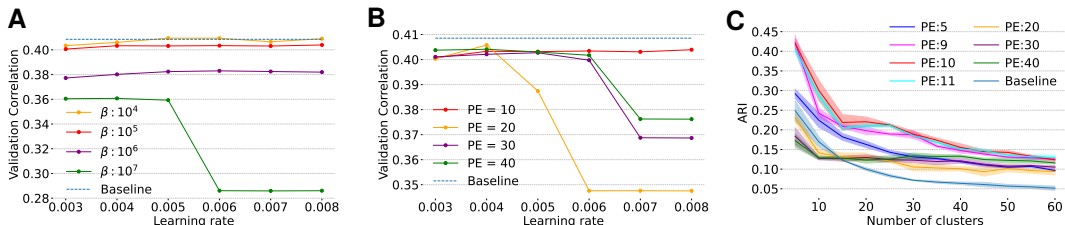

Figure 5: **A–B**: Learning rate tuning for $\beta$ (A) and length of pretraining (B). We fixed amount of clusters to 15. If the learning rate is too high the clustering loss starts oscillating due to learning rate scheduling leading to a massive drop in performance. **C** ARI for different number of pretraining epochs vs baseline model. For each number of pretraining epochs we used the optimal learning rate and set $\beta = 10^5$ for all experiments. All settings of DECEMber have better cluster structures after 15 clusters at latest. It is visible that 10 pretraining epochs generate the best clustered embeddings.

visual cortex remains unclear, with estimates ranging from 20 to 50 [20, 50], we considered a range of $j = 5, ..., 60$ in increments of 5.

We tested a wide range of loss strengths $\beta$, to find the optimal value to balance the clustering a nd the model loss. As $\beta$ increases, t-SNE vizualization suggests improved qualitative separation of clusters in the embedding space (Fig. 4B–E). However, this comes at a cost: when the clustering loss becomes dominant, the model's predictive performance drops significantly (Fig. 4F). This made us question if the qualitative structure in t-SNE plots is meaningful. To answer this question, we quantified clustering consistency using ARI between three model fits with different seeds and found that clustering consistency noticeably improved compared to the GMM baseline (Fig. 4H).

We see the ARI improvement as long as $\beta$ does not hurt performance ($\beta \leq 10^4$; Fig. 4G. However, once $\beta > 10^4$, performance starts suffering (Fig. 4F) and the ARI does not improve anymore (Fig. 4G), suggesting that the qualitative structure is created by removing functionally relevant heterogeneity between neurons. While ARI values double compared to the baseline model, there is no clear peak around a certain number of clusters. We would expect ARI to peak noticeably at the "true" number of clusters as shown for the retina ganglion cells (Fig. 3C) if such a structure existed. This suggests that mouse V1 likely lacks discrete functional cell types. Still, the clear improvement indicates meaningful local structure in functional embeddings.

**Consistency of embeddings depends on length of pretraining.** To validate our conclusions that mouse V1 lacks discrete functional cell types, we performed extensive tuning of DECEMber by using different numbers of pretraining epochs before turning on the clustering loss, different clustering strengths $\beta$ and tuned learning rates to optimize model predictive performance.

Across all settings, DECEMber achieved higher ARI scores than the baseline, indicating better consistency of embeddings (Fig. 4H, Fig. 5C).

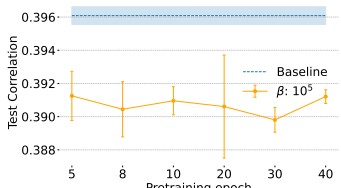

We found that the optimal learning rate varied depending on the number of pretraining epochs (Fig. 5A), and also depended on the clustering loss strength $\beta$ (Fig. 5B). Importantly, the choice of the number of pretraining epochs had minimal effect on the overall predictive performance if the learning rate was optimally chosen, with differences staying within the standard deviation across runs. We tuned on the validation set (Fig. 5B), and checked on the test set (Fig. 6). However, we observed a distinct peak in ARI for 10

Figure 6: The choice of pretrain epoch doesn't influence performance when we're using an optimal learning rate.

pretraining epochs in the case of the mouse visual cortex (Fig. 5C). While the ARI improved across a range of cluster settings, we did not observe a sharp maximum at any specific cluster count.

**DECEMber improves embeddings across different datasets and model architectures.** To ensure that DECEMber generalizes across architectures, modalities, and species, we additionally tested it on data from the mouse retina and macaque visual cortex area V4. We did not extensively tune hyperparameters, we only decreased the learning rate (lr) to stabilize the baseline model training for both datasets, and set $\beta$ as described in Sec. 4 (exact settings in App. A.15). More extensive tuning of the lr, $\beta$ or the number of pretraining epochs can lead to better results. For both datasets we preserved the performance of the original models (App. A.9).

For the mouse retina we used both the data and the models from Höfling et al. [18]. As for the marmoset retina, we trained a separate model for each retina to account for the temperature differences between retinas. Given the limited availability of cell-type labels, we included all cells in our analysis and evaluated cluster consistency across varying numbers of clusters. For details on dataset averaging and per-dataset analysis, see App. A.10.

For macaque V4 data we used spiking extracellular multi-electrode recorded responses of neurons to gray-scale natural images shown to awake macaque monkeys [51] and the model from Pierzchlewicz et al. [37], which had a different readout architecture – an attention readout instead of the previously used Gaussian readout. We trained the model on 1000 cells and measured the ARI across three model seeds.

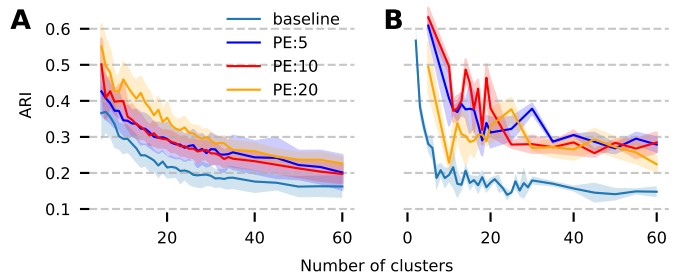

Figure 7: ARI on **A** mouse retina [18], weighted across six models. **B** monkeys V4 [51].

The embedding consistency doubled using our method (Fig. 7B). This shows that DECEMber is robust not only across different data modalities but also across architectures. For more analysis of the monkey dataset see App. A.11.

## 6 Discussion

In this work we introduced DECEMber, an additional training loss with explicit clustering bias for predictive models of neuronal responses. DECEMber enhances cluster consistency, while keeping state-of-the-art predictive performance. It is robust across different data modalities (electrophysiology and calcium imaging), species (mice, primates) and visual areas (retina, V1, V4). We also showed that DECEMber is robust across both static (mouse V1, macaque V4) and dynamic (retinas) cores and multiple readout architectures – the Gaussian readout and the attention readout.

We see DECEMber as a model-driven hypothesis test: if clear functional cell types exist, then incorporating this bias should improve model performance and/or the embedding structure, which we measure as cluster consistency. While improvements are observed across datasets and architectures, our main focus was mouse V1, where the existence of discrete excitatory cell types remains debated. Our results support the idea that excitatory neurons in mouse visual cortex form a functional continuum rather than discrete clusters. This finding is consistent with recent work studying different modalities by Weiler et al. [52], Tong et al. [19], and Weis et al. [6], who independently found no clear boundaries in morphological or electrophysiological features. In line with Zeng [1], we argue that future efforts to define mouse V1 cell types should emphasize multi-modality combining functional, morphological, and genetic data. This approach has proven fruitful in the retina, where functional types alone are coarser than those derived from multiple modalities [7, 21].

Given the generality of our clustering loss, which is model-agnostic and not tied to a specific architecture, we believe it holds promise for use in multi-modal models aiming to define cell types or in broader unsupervised representation learning contexts.

**Connection to other works learning neuronal embeddings.** There are other works [53–59], which all learn neuronal embeddings in some way, but none of them explicitly enforce or optimize for clustering, which is the main goal of DECEMber. Specifically, Nemo [53], NeurPIR [54] and NuCLR [55] are contrastive methods, DECEMber is not. Nemo [53] and NeurPIR [54] embeddings are functions of input (current activity, autocorrelogram), for DECEMber neuronal embeddings are time- and input- invariant weights of the predictive model (they embed the neuron's full input–output function). Nemo [53], NeuPRINT [56], NetFormer [57] do not model visual stimuli. NeuPRINT, NetFormer, POYO [58], NEDs [59] have time-invariant model weights, but both predict neuronal activity based on masked or previous neuronal activity while the regression model in our paper predicts neuronal activity based on visual stimuli. It might be interesting to integrate DECEMber clustering loss with these works [56–59] but we leave it for future work.

**Limitations.** DECEMber requires a predefined number of clusters. When this is unknown, multiple runs with varying cluster counts and seeds are necessary in combination with an evaluation ARI-like metric to identify the optimal configuration. This increases the computational cost. Choosing an appropriate clustering strength $\beta$ is also crucial for balancing ARI and model performance and further work is needed to determine the optimal pretraining duration.

Moreover, operating in high-dimensional feature spaces introduces an additional challenge: the cluster covariance matrices can become large and ill-conditioned, with tiny diagonal values, hitting the limits of numerical stability. We address this issue by clamping small values, though this solution is heuristic rather than principled. Furthermore, high-dimensional settings require a sufficient number of data points to prevent overfitting of the scale matrices. Additionally, we currently assume a $t$-distributed feature space via a $t$-mixture model, but this can be adjusted if a more suitable prior over the embeddings is known.

# 7 Acknowledgments

We thank Suhas Shrinivasan, Max F. Burg, Larissa Höfling, Thomas Zenkel, Konstantin F Willeke, Fabian Sinz.

This work was supported by the Deutsche Forschungsgemeinschaft (DFG, German Research Foundation) – project IDs 432680300 (SFB 1456, project B05) and 515774656 – and by European Research Council (ERC) under the European Union's Horizon Europe research and innovation programme (Grant agreement No. 101041669). AST acknowledges support from National Institute of Mental Health and National Institute of Neurological Disorders And Stroke under Award Number U19MH114830 and National Eye Institute award numbers R01 EY026927 and Core Grant for Vision Research T32-EY-002520-37. We gratefully acknowledge the computing time granted by the Resource Allocation Board and provided on the supercomputer Emmy/Grete at NHR-Nord@Göttingen as part of the NHR infrastructure. The calculations for this research were conducted with computing resources under the projects nim00010 and nim00012.

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

# A   Appendix

## A.1   Signal to noise ratio for toy example

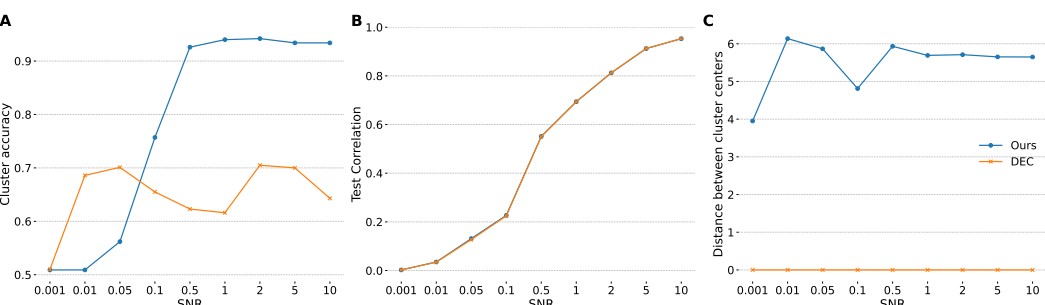

Figure 8: **A**: Clustering accuracy for predicted vs. ground-truth label for DEC vs. DECEMber for different Signal-to-Noise ratios. **B**: Predictive performance on test set. **C**: The distance of the norms of the 2 cluster centroids $\|\mu_1\| - \|\mu_2\|$ for DEC vs. DECEMber. The ground truth distance is $\sqrt{128 \cdot 1.5^2} - \sqrt{128} \approx 5.66$.

Data in mouse V1 is recorded using 2-photon calcium imaging which is known to be noisy. To investigate how robust DECEMber is towords noise we varied the SNR in the toy example setting. As described in the toy example we generated clean responses as the dot product of the stimuli and the created network weights. To simulate noisy observations, we added Gaussian noise independently for each neuron and stimulus but with the same variance. The noisy responses are thus given by $y_{ij} = z_i^T x_j + \epsilon_{ij}$, where $z_i$ are the neurons' weights, $x_j$ the stimuli and $\epsilon_{ij}$ Gaussian noise. Since both the input and the noise are mean-centered, the Signal-to-Noise Ratio of the neuronal population of $N$ neurons and $M$ stimuli is defined as $SNR = \frac{1}{NM} \sum_{i=1}^{N} \sum_{j=1}^{M} \frac{\mathrm{Var}\left[z_i^T x_j\right]}{\mathrm{Var}\left[\epsilon_{ij}\right]}$.

In this setup, we vary the SNR by adjusting the noise variance, thereby controlling the noise power in the simulation. We varied the SNR between 0.001 and 10 and trained in the same way as before. We calculated the Pearson correlation on a left out test set. Both DEC and DECEMber converge to similar performance since the MSE drives the learning of the model's weights (Fig. Fig. 8B). We see that even for low SNR DECEMber successfully separates the clusters (Fig. Fig. 8A), though the distance is not ideal before SNR is above 0.1. However, for DEC cluster collapse happens independently of SNR (Fig. Fig. 8C). Since the weights (e.g. neurons) are generated based on predefined clusters (means of the Gaussians), we know the ground truth label for each neuron. To evaluate the cluster accuracy of DEC and DECEMber we measure the proportion of neurons that are assigned to the correct cluster.

## A.2   Retina gagnlion cells

To select reliable cells from the marmoset RGCs dataset [48], we used the same reliability assessment of each cell's responses to visual stimuli as in [60]; only reliable cells were used for model training. The model architecture was also taken from [15]. For clustering evaluations using DECEMber, k-means, and GMM, we considered only cells for which cell-type labels were available.

The dataset contains recordings from two different retinas of male marmosets. The second retina (not analyzed in the main part of this paper) includes 38 parasol-OFF and 35 parasol-ON cells which are well separable (Fig. 9B). We trained our models on all reliable cells from

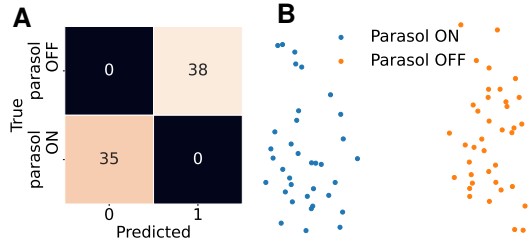

Figure 9: **A**: DECEMber predictions. Pretraining length: 20 epochs. Same predictions for GMM and k-means. All methods have ARI 1. **B**: $t$-SNE projections of the corresponding cells.

this retina as well and tested DECEMber with
varying pretraining lengths (which did not affect cluster consistency). All three clustering meth-
ods—GMM, k-means, and DECEMber—successfully and robustly identified the two cell types, as
visualized in Fig. 9A with ARI=1.

**Cell type labels**    We used the same cell-type classification procedure as in [60] (Methods section
4.5), clustering the cells using the KMeans++ algorithm on features extracted from receptive-field
estimates obtained using spike-triggered averaging from responses to spatiotemporal white-noise, and
from autocorrelograms computed on responses to white-noise and naturalistic movies. The cell-type
labels in our analysis differ from the ones used in the original publication because we did not exclude
cells that violated the tiling of spatial receptive fields.

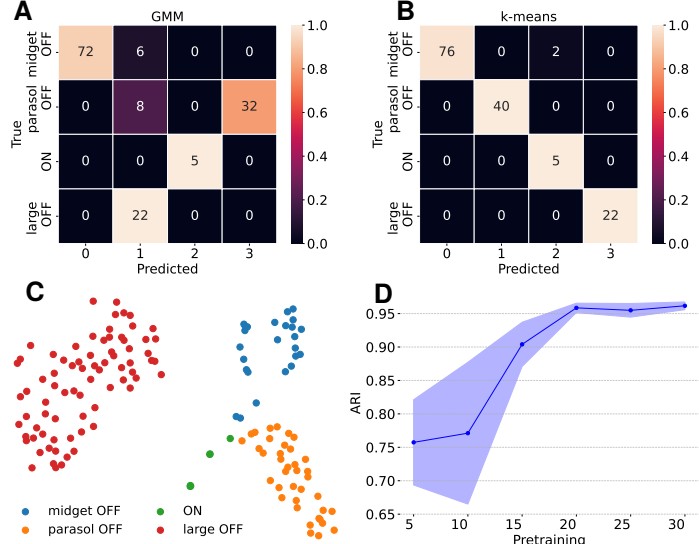

Figure 10: All plots show evaluations of seed 4 of the trained marmoset RGC model [15] for retina
1 used in the main part of this paper. **A**: GMM predictions. **B**: k-means. **C**: $t$-SNE projections of
the corresponding cells. **D**: ARI for different length of pretraining. Longer pretraining seems to be
beneficial with ARI stabilizing after pretraining of 20 epochs.

### A.3  ARI-stability for k-means and GMM on marmoset RGC

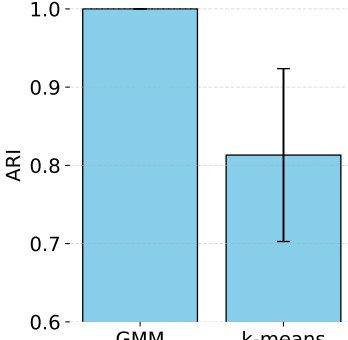

Figure 11: **Clustering stability of k-means and
GMM on retina 1 for marmoset RGC.** We started
with a single baseline RGC model of retina 1 (seed
2) and performed k-means and GMM clustering (4
clusters each), varying the random seed (42, 10, 100)
for both algorithms. Clustering was done on all cells
the model was trained on, but ARI was calculated
using only labeled cells.

## A.4 ARI stability for GMM on mouse V1

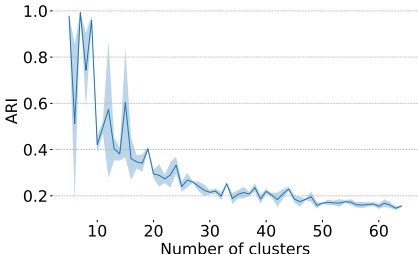

Figure 12: **Clustering stability of GMM on mouse V1.** We trained a baseline model of mouse V1 for one seed (42). We then did GMM for clusters ranging from 5 to 60 as the ground truth is not known varying just the seed for the initialization of the GMM. It's clearly visible that GMM becomes unstable if the amount of clusters is large.

## A.5 Sensorium data details.

The model was trained on the SENSORIUM 2022 dataset [8], which contains neural responses to natural images recorded from seven mice (a total of 54,569 neurons). Recordings were made from excitatory neurons in layer 2 and 3 of the primary visual cortex using two-photon calcium imaging. In addition to neural activity, the dataset includes five behavioral variables: locomotion speed, pupil size, the instantaneous change in pupil size (estimated via second-order central differences), and horizontal and vertical eye position, all of which are incorporated into the model. Three of them – locomotion speed, pupil size, and the instantaneous change in pupil size – were appended to the grayscale images and are used as input to the core, while pupil horizontal and vertical position were used as input to the shifter – a model part shifting the readout receptive field locations depending on where the mouse is looking. The validation set contains responses to roughly 500 and test set to 5000 images per mouse.

## A.6 Additional clustering metrics show qualitatively consistent results with ARI

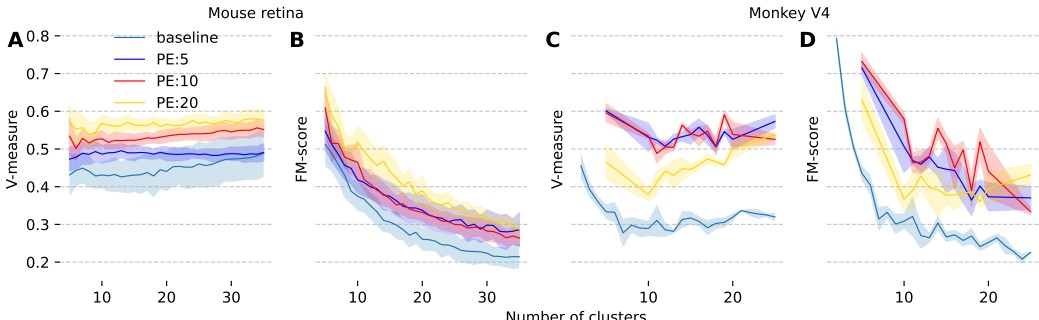

Figure 13: Different clustering consistency metrics for monkey V4 and mouse retina datasets. Same as in the main paper, mouse retina is weighted across six models using all neurons, monkey V4 model is trained on a subset of 1000 neurons. The order of lines is same as for ARI, confirming its results qualitatively. V-measure is biased towards bigger amount of clusters due to the set-based nature.

Clustering quality can be evaluated using metrics beyond ARI. ARI measures the similarity between two clusterings by checking whether pairs of points are assigned to the same cluster in both. The Fowlkes-Mallows index [61] (Fig. 14) also compares two partitions but does not adjust for chance; it is the geometric mean of precision and recall, based on how consistently point pairs are clustered together.

Other common metrics – homogeneity, completeness, and V-measure [62] – are asymmetric and compare one clustering against a reference (typically ground truth). Homogeneity measures whether each cluster contains only members of a single class, while completeness checks whether all members of a given class are assigned to the same cluster. Swapping the roles of predicted and true labels interchanges homogeneity and completeness. V-measure, equivalent to normalized mutual information (NMI [63]) and it is the harmonic mean of the two. As in our case we do not have ground truth, we

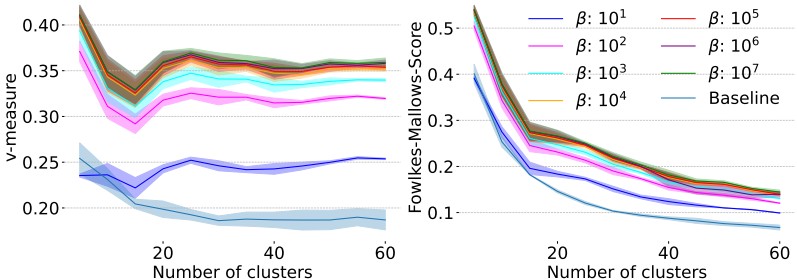

Figure 14: V-measure and Fowlkes-Mallows-score for PE 10, 15 clusters.

compute the metrics with all possible seed pairs, which leads to homogeneity, completeness, and V-measure being equivalent (Fig. 14).

## A.7 Influence of Degree of freedom as a hyperparameter

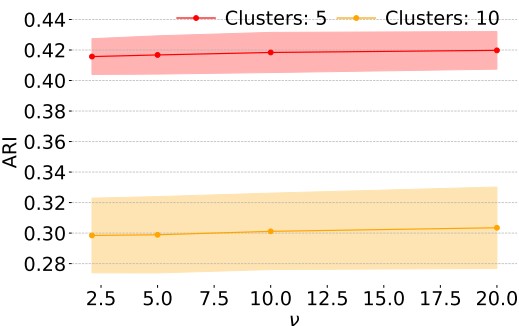

Figure 15: DECEMber, with PE = 10, lr=0.008, $\beta = 10^4$ with different degrees of freedom $\nu$.

We've tested the influence of the degree of freedom on ARI for both 5 and 10 clusters 15 and can't really see a difference.

## A.8 Comparison with the rotation equivariant baseline

Turishcheva et al. [13] is the only work to date that specifically addresses neuronal embedding consistency, and thus serves as our baseline for comparison. We use the $\gamma_{\text{lognorm}} = 10$ condition from their paper and compare it to our consistency results in Fig. 16. Our approach achieves comparable consistency levels while eliminating the need for a rotation-equivariant core, thereby removing the post-hoc alignment step and improving predictive performance from $\approx 38.1\%$ (Fig. 3 A in [13]) to $\approx 39.5\%$ (Fig. 4F).

## A.9 Models performances on mouse retina and macaque V4 data

|  | Baseline | PE 5 | PE 10 | PE 20 |
|---|---|---|---|---|
| Mouse retina | $0.4727 \pm 0.0008$ | $0.4695 \pm 0.0009$ | $0.4732 \pm 0.0008$ | $0.4727 \pm 0.0009$ |
| Macaque V4 | $0.308 \pm 0.004$ | $0.308 \pm 0.006$ | $0.304 \pm 0.005$ | $0.305 \pm 0.003$ |

Table 1: Performances on mouse retina and macaque V4 data for the models reported in the main paper (Sec. 5). Mouse retina is weighted as described in App. A.10.All performances are on the held-out test set. The values are averaged across all cluster counts. Seeds were 42, 101 and 7607. For GMM baseline the seed was 42.

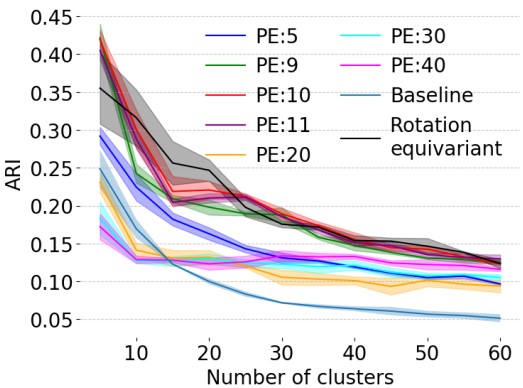

Figure 16: DECEMber, with PE = 10; DECEMber cluster consitency matches the rotation-equivariant model from Turishcheva et al. [13].

## A.10 Further analysis of mouse retina data

**Averaging across datasets** For the mean ARI across datasets we weighted ARI lines like $\mu_{\text{total}} = \sum_i w_i \mu_i$, where $w_i = n_{\text{cur}}/n_{\text{total}}$ with $n_{\text{cur}}$ - the number of neurons in the current model, $n_{\text{total}}$ is the number of neurons in all six models, and $\mu_i$ is the average ARI score across three seeds for the current model. We used the law of total variance and computed the variance as $\sigma^2_{\text{total}} = \sum_i w_i \left[ \sigma_i^2 + (\mu_i - \mu_{\text{total}})^2 \right]$, where the first term captures within-dataset ARI variability and the second term captures between-dataset ARI variability. Fig. 17 shows the ARIs per models. We can see that the fewer neurons were present in the models the less the improvement was.

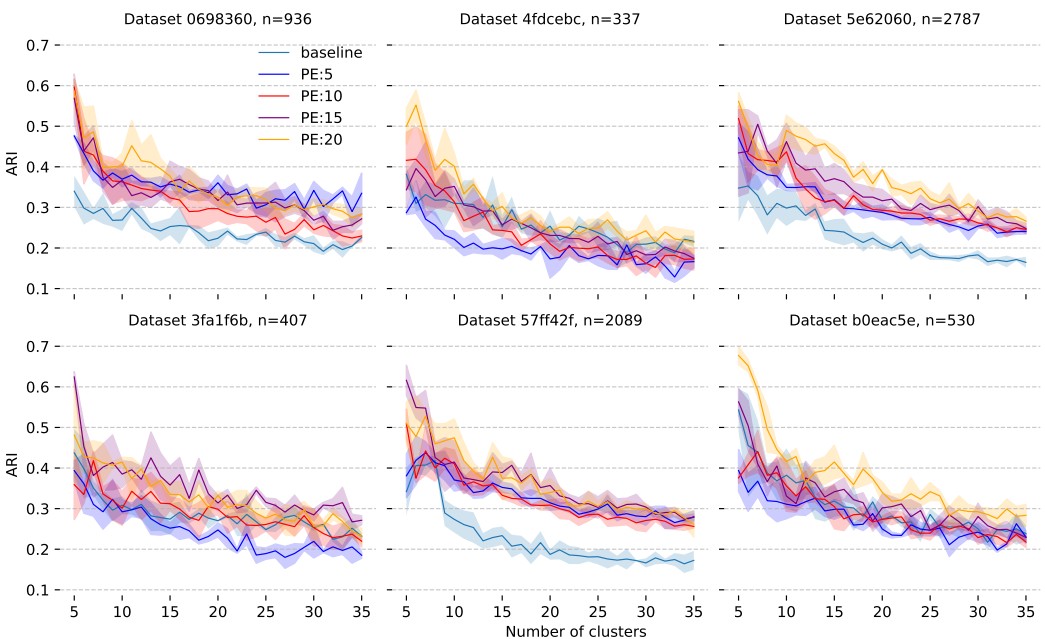

Figure 17: ARI per retina. $n$ is the number of neurons in the model

## A.11 Further analysis of monkeys data

For monkey V4, we trained models for 5, 10 to 20 and 25 clusters, as original work reported 12 clusters for 1000 cells. For 144 cells there were no labels and 100 cells ahd a "not properly clustered" label. Therefore, we decided to use only the 1000 labeled cells. For results of models trained on all

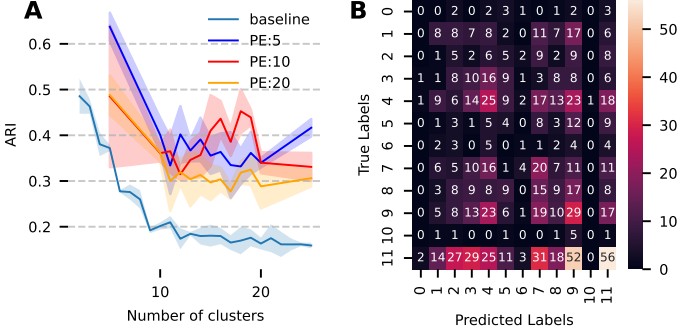

Figure 18: **A** ARI between models seeds for models trained on all 1244 neurons.
**B** Confusion matrix between predictions of the model trained on 1000 neurons and labels suggested in Willeke et al. [51]

cells see Fig. 18A. While the trends and values are qualitatively similar to the model trained only on a 1000 neurons subset, the standard deviation corridor seems to be wider, likely due to some of the "not properly clustered" neurons being in between the distinct groups. Please note that the labels from Willeke et al. [51] are rather a suggestion but not ground truth as they were not verified using independent biological measurements. For the confusion matrix of our labels and labels from Willeke et al. [51] see Fig. 18A. Same as for Burg et al. [21], our labels do not perfectly match the ones proposed in Willeke et al. [51].

## A.12 Bayesian inference criterion BIC

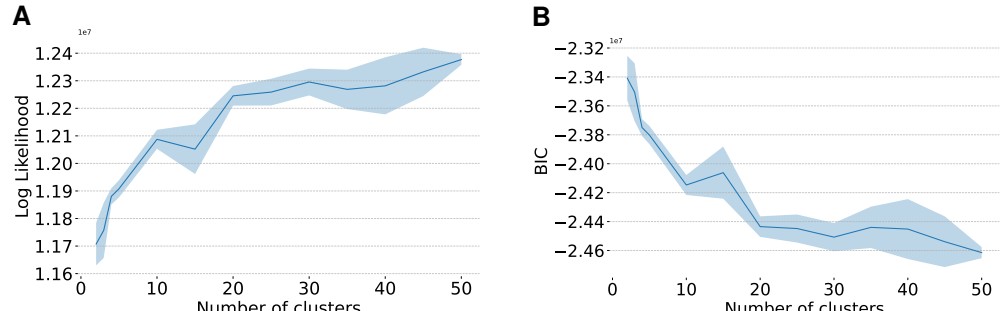

Figure 19: **A**: Log likelihood estimation. **B**: Calculation of BIC.

As suggested by a reviewer, BIC could be an alternative way to find the underlying number of clusters. We used a model with 10 pretraining epochs and $\beta = 10^4$.

After training the model, we calculated the log-likelihood of the neurons

$$\log \mathcal{L} = \sum_{i=1}^{N} \log \frac{1}{K} \sum_{k=1}^{K} t_\nu(\mathbf{x}_i \mid \boldsymbol{\mu}_k, \boldsymbol{\Sigma}_k),$$

where $t_\nu(\cdot \mid \boldsymbol{\mu}_k, \boldsymbol{\Sigma}_k)$ is the multivariate $t$-distribution with mean $\boldsymbol{\mu}_k$, covariance $\boldsymbol{\Sigma}_k$, and degrees of freedom $\nu$.

We computed the BIC as $\mathrm{BIC} = -2 \log \mathcal{L} + (2 * K * d + 1) \log(N)$,

where $N$ is the number of neurons, $d$ is the dimensionality of the embeddings, $K$ is the number of clusters.

The likelihood grows with the number of clusters whereas BIC falls with the number of clusters (Fig Fig. 19) indicating the lack of a clear cluster peak, which agrees with ARI (Fig Fig. 4H). BIC also scales with the number of model parameters which in our case increase a lot when the number of clusters increases making it not the most suitable measure in our case.

### A.13 Compute requirements

All of our models can be considered light-weight in terms of compute by modern deep learning model standards. A single mouse retina model requires less the 10Gb of GPU memory and trains under 20 minutes of walltime. A single mouse V1 model requires $\approx$ 12Gb of GPU memory and trains for under 2 hours of walltime. A single marmoset retina model uses 40Gb GPU and trains for under 16 hours of walltime. A single monkey model requires 24Gb of memory and trains for under 2 hours of walltime.

We use a local infrastructure cluster with 8 NVIDIA RTX A5000 GPUs with 24Gb of memory each for mouse experiments. For mouse retina, marmoset retina, and monkey V4 we used 40Gb NVIDIA A100.

### A.14 Broader impact

Our work contributes to building more reproducible models, which are more suitable for making biologically meaningful statements. It is even more related to derive a functional taxonomy of cell types in the primary visual cortex, which can enhance our understanding of brain function and support the development of treatments for neurodegenerative diseases.

### A.15 Experimental settings

For marmoset RGC dataset, we used the three layer CNN described in [15]. We trained it for a maximum of 1000 epochs, stopping early if validation correlation did not improve for 20 epochs. The learning rate of both pretraining and training with the clustering loss was initially 0.005 and reduced during training using the `ReduceLROnPlateau` learning rate scheduler, patience 10 and minimal learning rate $1e^{-8}$.

For SENSORIUM 2022, we used their model and training hyperparameters for the baselines training. Pretraining duration, learning rates and clustering strength $\beta$ is reported in every experiment. For mouse retina, we followed Hofling et al. [18] model and training hyperparameters, changing only learning rate from 0.01 to 0.005 to improve baselines stability. Clustering strength was set to 0.001 across all experiments. For monkey V4 data we followed model and training hyperparameters from [37], again only changing the learning rate from $3 \cdot 10^{-4}$ to $5 \cdot 10^{-5}$ to improve baselines stability. Clustering strength was set to 0.001 across all experiments. Changing learning rate in both cases did not impacted performance in more than std boundaries.

