# OpenReview forum: "Learning to cluster neuronal function"
_NeurIPS.cc/2025/Conference — NeurIPS 2025 poster_

### Official Review · Reviewer_gTJC · 2025-06-24

**Clarity:** 4
**Significance:** 4
**Originality:** 4
**Rating:** 6
**Confidence:** 3

**Summary:**

This paper points out a problem in neural activity predictive models: Although they can achieve high predictive performance, the extracted features do not exhibit clear clustering patterns that correspond to cell types. This paper then proposes to add a clustering loss to the neural activity prediction objective to explicitly enforce clustering. The clustering loss is also a novel one that improves the existing deep embedding clustering loss, which failed to separate a toy dataset. Then the paper goes on to validate their method across 4 datasets (retinal ganglion cells, mouse v1, mouse retina, and monkey v4) and demonstrates a good generalization across species and visual areas. By using their method on mouse v1, the authors conclude that mouse v1 cell types may not have distinct functional types, which aligns with other findings in literature.

**Questions:**

This is overall a very strong paper, and I do not have many questions.

1. May I suggest an additional experiment on varying the number of clusters (k ranging from 1 - 10, for example) on the retinal ganglion cell dataset? Plot ARI to see if it peaks at the true number of types. This would demonstrate that the method can recover a correct cluster count when discrete classes exist. This experiment would further support your claim that “We would expect ARI to peak noticeably at the ‘true’ number of clusters if such a structure existed. This suggests that mouse V1 likely lacks discrete functional cell types.”
2. Do you think your method can be improved so that it can handle data across multiple subjects or sessions?

**Ethical Concerns:**

["NO or VERY MINOR ethics concerns only"]

**Final Justification:**

I appreciate that the authors have conducted an additional experiment on the retina data to demonstrate the validity of their approach. Overall, the authors have done thorough experiments and analyses, and I am particularly excited that the proposed method can lead to the conclusion that mouse v1 neurons may not have functional subtypes. I think this is a novel and meaningful contribution to the research community.

**Limitations:**

yes

**Quality:**

4

**Strengths And Weaknesses:**

Strengths:
* The paper is well-written, easy to follow, and well-motivated.
* The paper proposes a novel clustering loss to address that the vanilla DEC loss fails to separate a simple toy dataset
* The paper's algorithm is validated on a wide range of datasets, across species and visual areas. I like that the authors first validate the algorithm with a toy dataset for a sanity check, then go on to real datasets in increasing complexity (retinal ganglion cells, mouse v1, mouse retina, and monkey v4).

Weaknesses:
* As the authors have already discussed in the Limitations section of the paper, this algorithm requires tuning of different hyperparameters, including the number of clusters, the strength of the clustering loss, and the number of pertaining epochs. This could mean the algorithm is hard to use in practice.
* The clustering algorithm does not seem to be able to handle data from different subjects/trials altogether.

---

> ### Author Rebuttal · Authors · 2025-07-31
>
> Thank you for appreciating our paper, noticing novelty and validation *“on a wide range of datasets, across species and visual areas”* and toy data.
>
> We are happy to address you comments and questions below:
> ### **Questions**
> **Q1**: *Additional experiment on marmoset retina*: Thanks a lot for suggesting an additional experiment with varying the number of clusters on the retinal ganglion cell dataset. The table below reports ARI values per cluster (and std across 3 seeds). We indeed observe a clear peak around 4 clusters, as it should be for this retina (Fig.3 in the original manuscript), so ARI does “peak noticeably at the ‘true’ number of clusters if such a structure exists.”
> We will add this  experiment as a figure into the final version of the paper.
>
> | Clusters | ARI (mean ± std) |
> |----------|------------------|
> | 2        | 0.78 ± 0.07      |
> | 3        | 0.52 ± 0.31      |
> | 4        | **0.95 ± 0.01** |
> | 5        | 0.78 ± 0.08      |
> | 6        | 0.77 ± 0.15      |
> | 7        | 0.81 ± 0.08      |
> | 8        | 0.85 ± 0.05      |
> | 9        | 0.79 ± 0.04      |
> | 10       | 0.85 ± 0.07      |
> | 15       | 0.71 ± 0.07      |
> | 20       | 0.74 ± 0.06      |
> | 25       | 0.67 ± 0.04      |
> | 30       | 0.49 ± 0.02      |
> | 40       | 0.41 ± 0.04      |
>
> **Q2**: *Multiple sessions*: Actually, our method is capable of handling data from different subjects/trials altogether, in fact, it is already done for mouse visual cortex and macaque V4. We do indeed handle sessions separately for mouse and marmoset retinas due to the differences in the experimental conditions (presumably different temperature) which influences the speed of the responses so much that this would be the most trivial dimension to separate the neurons for different groups [1,2]. While several works try to address this issue and integrate several retinas into the same model [3,4,5], this is still an active area of research, and is not the main focus of our paper.
>
> ### **Weaknesses**
> **W1** *Hyperparameters*: Our method indeed introduces additional hyperparameters, but not all of them need extensive tuning. Specifically, β just scales the clustering loss to the same order as the Poisson loss, which can be done with 1-2 short runs. However, finding the optimal pretraining epoch or number of clusters indeed increases the amount of necessary experiments. But since there are no competing methods which achieve the same without hyperparameter tuning, we believe that this degree of tuning is manageable and not unusual.
>
> **W2** *Multiple sessions*: See **Q2**
>
> We thank you once again for constructive feedback and helping us to improve our paper. We are happy to answer any follow-up questions.
>
> **References**:
> [1] Zhao, Z., Klindt, D. A., Maia Chagas, A., Szatko, K. P., Rogerson, L., Protti, D. A., ... & Euler, T. (2020). The temporal structure of the inner retina at a single glance. Scientific reports, 10(1), 4399.
> [2] Fritsches, K. A., Brill, R. W., & Warrant, E. J. (2005). Warm eyes provide superior vision in swordfishes. Current Biology, 15(1), 55-58.
> [3] Höfling, L., Szatko, K. P., Behrens, C., Deng, Y., Qiu, Y., Klindt, D. A., ... & Euler, T. (2024). A chromatic feature detector in the retina signals visual context changes. Elife, 13, e86860.
> [4] Vystrčilová, M., Sridhar, S., Burg, M. F., Gollisch, T., & Ecker, A. S. (2024). Convolutional neural network models of the primate retina reveal adaptation to natural stimulus statistics. bioRxiv, 2024-03.
> [5] Gonschorek, D., Höfling, L., Szatko, K. P., Franke, K., Schubert, T., Dunn, B., ... & Euler, T. (2021). Removing inter-experimental variability from functional data in systems neuroscience. Advances in Neural Information Processing Systems, 34, 3706-3719

---

> > ### Comment · Reviewer_gTJC · 2025-08-01
> >
> > Thank you for conducting the additional experiment and addressing my questions. I do not have further questions at this time.

---

### Official Review · Reviewer_2CDw · 2025-07-01

**Clarity:** 2
**Significance:** 2
**Originality:** 2
**Rating:** 3
**Confidence:** 3

**Summary:**

This paper addresses a core question in computational neuroscience: whether neuronal functions constitute discrete cell types, particularly in the mouse primary visual cortex (V1). Existing methods cluster neuron embeddings learned by predictive models, but the results of such clustering lack consistency. The authors propose a new method called DECEMber, which introduces an explicit clustering inductive bias by adding an auxiliary clustering loss, inspired by deep embedding clustering, to the predictive model's loss function. The core innovation of this method is its use of a multivariate Student's t-mixture model combined with the Expectation-Maximization (EM) algorithm to update clustering parameters, thereby adapting to the non-isotropic features of neuronal embeddings.

The authors validate the method's effectiveness on a "positive control" dataset where discrete cell types are known to exist—marmoset retinal ganglion cells—and apply it to several other datasets, including mouse V1, mouse retina, and macaque V4. The main conclusion is that although DECEMber improves clustering consistency, it does not find strong evidence supporting the existence of discrete functional cell types in mouse V1, suggesting its functional organization is more likely a continuum.

**Questions:**

see  Weaknesses

**Ethical Concerns:**

["NO or VERY MINOR ethics concerns only"]

**Final Justification:**

The innovation of this article is limited, as many existing works have achieved similar results, and the author has not made a comprehensive and fair comparison. So I choose to reject it.

**Limitations:**

yes

**Quality:**

2

**Strengths And Weaknesses:**

Strengths：
(1)Importance of the Problem: The paper investigates a fundamental and challenging problem in neuroscience—the identification of functional cell types. Applying machine learning methods to test this biological hypothesis holds significant scientific value.

(2)Clear, Hypothesis-Driven Framework: Structuring the method as a form of "model-driven hypothesis testing" is a clever and rigorous approach. That is: if discrete functional types do exist, introducing a clustering bias should improve certain performance metrics of the model.

(3)"Positive Control" Experimental Design: The experimental results on the marmoset retina dataset, known to have discrete cell types, are convincing. DECEMber achieved an extremely high ARI score (0.96) and near-perfect classification results on this dataset (Fig. 3B), which strongly demonstrates the method's potential under ideal conditions.

(4)Exhaustive Experiments and Transparency: The authors conducted extensive experiments across multiple species (mouse, primate), multiple brain regions (retina, V1, V4), and multiple model architectures (Gaussian readout, attention readout). Furthermore, the paper is clearly written and candidly lists several limitations of the method in the discussion section.

3.Weaknesses
Despite the aforementioned strengths, there are several significant weaknesses that make this paper unsuitable for publication at NeurIPS.

(1)Limited Novelty: The core of the method is combining a clustering loss with a prediction loss. This idea is not new. DECEMber is essentially a combination of existing techniques: Deep Embedding Clustering (DEC, 2016), t-mixture models (TMM, 2000), and the EM algorithm. While its application in the field of neuroscience is new, from a machine learning methodology perspective, its conceptual contribution is incremental and does not meet the high standards of NeurIPS.

(2)Complex Method and Highly Sensitive to Hyperparameters: DECEMber introduces a large number of key hyperparameters that require careful tuning, including the clustering loss weight β, the number of pre-training epochs m, the number of clusters J, and the learning rate. One of the paper's core claims is that the "model improves clustering consistency while maintaining high predictive performance". However, Figure 4F clearly shows this is a direct trade-off, not maintenance. When the β value is increased to obtain better t-SNE visualizations, the model's predictive performance drops significantly. Performance is only unaffected at small β values (≤10⁴), but at that point, the improvement in consistency is also limited.

(3)Tedious Tuning Process: Figures 5A and 5B show that the model's performance and consistency are sensitive to the choice of learning rate and the number of pre-training epochs when trying to achieve optimal parameters. For example, on the mouse V1 data, the ARI peaks at 10 pre-training epochs. This means users need to perform an expensive grid search to find the optimal hyperparameter combination for each new task, which greatly undermines the method's practicality and robustness. The authors themselves admit that selecting an appropriate β is crucial and that the pre-training duration requires "further work" to determine.

(4)The Method's Value Is Not Realized: For mouse V1, the paper's main scientific conclusion is that discrete functional cell types "likely do not exist," but rather form a continuum. While this is a valuable scientific finding, it fundamentally undermines the motivation for proposing this complex clustering method. If the core problem the method aims to solve does not have a clustered solution, what is the point of introducing this complex clustering mechanism? This makes the primary application scenario of the method seem somewhat contradictory.

(5)Methodological Flaw: The authors acknowledge that to address potential numerical stability issues with high-dimensional covariance matrices, they adopted a "heuristic rather than principled" approach of truncating smaller values. Relying on such an ad-hoc solution in a core algorithmic component is a significant methodological flaw.

(6)Inconsistent and Potentially Misleading Baseline Comparison: A core argument of the paper is that DECEMber outperforms standard methods like GMM and k-means in clustering stability. However, the evidence for this claim is contradictory. In the main results (Fig. 3A), the GMM baseline has an ARI of about 0.85 and shows huge error bars, indicating its instability across different model seeds. However, in the appendix (Fig. 10), the authors show that on the embeddings from a single pre-trained model, GMM clustering itself is highly stable (ARI ≈ 1.0), while k-means is unstable. The discrepancy between these two results is not explained in the text. This strongly suggests that the instability of GMM in Figure 3A stems mainly from the differences in the embedding spaces learned by the predictive model itself under different random seeds, not from the GMM algorithm itself. If so, comparing an end-to-end trained method like DECEMber with a post-hoc GMM clustering on unstable embeddings is unfair. This inconsistent presentation severely weakens the core claim that DECEMber is superior to the baselines.

(7)Poor Data Visualization Hinders Understanding of Core Results: The presentation of several figures in the paper is flawed, making it difficult for the reviewer to evaluate its key arguments.

①Fragmented Presentation of the Core Trade-off in Figure 4: Figure 4 is central to the paper's argument regarding the mouse V1 results. However, the reader must jump back and forth between subplot 4F (predictive performance) and subplot 4H (clustering consistency ARI) to understand the trade-off between performance and consistency. A more direct and convincing visualization would be to plot these two key metrics on the same graph (e.g., x-axis for ARI, y-axis for predictive performance, with different colors/symbols for β values). The current fragmented presentation obscures rather than clarifies the core issue.

②Coarse and Inconsistent Presentation of Generalization in Figure 7: The authors claim that DECEMber has good generalization properties, but the presentation in Figure 7 fails to support this claim convincingly.

③Overly Brief Figure Caption: The caption for Figure 7 is extremely uninformative, only stating the dataset sources. It does not explain the trends or key findings in the figure, leaving the reader unable to understand its intent independently.

④Inconsistent Evaluation Criteria: The x-axis ranges (number of clusters) in Figure 7A (mouse retina) and Figure 7B (macaque V4) are different. More importantly, their evaluation methods differ: A is a weighted average across 6 models, while B is a simple average across 3 model seeds. This lack of a unified evaluation framework makes cross-dataset comparison meaningless and severely undermines the claim of "generalization".

(8)Ambiguous Contribution to the Core Scientific Question: The authors position this method as a tool to test for the existence of discrete functional cell types in mouse V1. However, even with DECEMber, the best ARI value obtained on mouse V1 is still very low (around 0.25). While this is a relative improvement over the baseline (around 0.15), an absolute value so close to random (ARI=0) can hardly be considered a "meaningful" cluster structure. Therefore, concluding that "mouse V1 likely lacks discrete functional cell types" based on an improvement from 0.15 to 0.25 is a weak argument. The method's improvement is too small to provide strong evidence for answering this important biological question.

---

> ### Author Rebuttal · Authors · 2025-07-31
>
> Thank you for your thoughtful feedback, highlighting that we are solving an important problem with a *"clear, hypothesis-driven framework"*.
> Below we address your specific comments from the review:
>
> *(1) Limited Novelty*:
> Of course our method builds on prior work (DEC, EM, TMM), but it introduces key innovations:
> The novelty is the combination of solving a multivariate non-linear regression task (how neurons respond to images) jointly with clustering the neurons’ readout weights. Unlike DEC, which clusters input samples by altering the model’s latent space and enforcing unit-scale clusters, we cluster model weights directly, constrained by the Poisson loss, which is jointly optimized with the clustering loss. This requires significant adaptation, including introducing EM updates, learning the scale matrix and an appropriate curriculum of first pre-training for the regression task. DEC serves more as inspiration than as a direct combination of methods.
> Our model provides a hypothesis test for a core neuroscience question—whether distinct functional cell types exist in mouse V1. Currently no such method exists!
> The components may be known, but their integration, adaptation, and application are novel and scientifically meaningful.
>
> *(2) Method Complexity*:
> DECEMber has several hyperparameters, but not all need tuning. For example, β just needs to roughly balance clustering and Poisson losses, which can be estimated in 1–2 few epochs runs. In several experiments (marmoset, mouse retina, monkey V4), we did exactly this, without sweeping β.
> Regarding Figure 4: It is correct that very large β>=10^5 hurt performance (Fig. 4F). However, consistency improves substantially compared to the baseline already with much smaller β~=10^4 (Fig. 4H), in which case the high predictive performance is maintained. Thus, **there is no trade-off**. We will improve the figures to make this clearer in the final version.
> The experiment of increasing the clustering weight, β, to the point where performance drops is intentional. It demonstrates that aggressively forcing a clustered structure can erase functionally relevant details, but does not improve embeddings’ consistency. This result supports our conclusion that excitatory neurons in the mouse visual cortex likely form a continuum rather than discrete types. **It is not a weakness of the method – it is a property of the data!**
> Finally, t-SNE plots were used only for visualization, not for tuning β—we’ll make this clearer in the text.
>
> *(3) Tedious Tuning*:
> You're right that careful hyperparameter selection matters and we appreciate the opportunity to clarify DECEMber’s practicality and robustness.
> We believe that the degree of tuning required is manageable. **Even sub-optimal hyperparameters yield substantial benefits**. The ARI for mouse V1 data does peak at 10 pretraining epochs, but Fig. 5C shows that nearly all tested settings outperform the baseline across a range of cluster counts. This suggests DECEMber offers more consistent embeddings than standard methods, even without a perfectly tuned grid search.
> On other datasets (mouse retina and macaque V4), we explicitly avoided extensive tuning—only making minor learning rate adjustments for stability and choosing β as described above (1 additional run to estimate clustering loss order) and 3 different pretraining epochs to try out—and still observed a doubling of ARI without compromising predictive accuracy (see App. A6 for predictive performances). This highlights DECEMber's robustness and strong out-of-the-box performance.
> We acknowledged these limitations in our discussion. As with many flexible deep learning methods, some tuning is expected, but the resulting gains in embedding stability and interpretability justify the effort.
>
> *(4) Value of the Method:*
> You ask “*If the core problem the method aims to solve does not have a clustered solution, what is the point of introducing this complex clustering mechanism?”* – The point is that **it is an open scientific question whether the solution is clustered**. Neurons are widely thought to form discrete types morphologically and genetically, and in the retina, this holds for function as well [6]. But whether V1 neurons cluster functionally is unknown. Our method tests this: if discrete types exist, clustering should improve model structure or performance (lines 314–316). This holds in the retina (Fig. 3), where functional types are known.
> Moreover, as reviewer gTJC suggested, varying cluster count for marmoset retina reveals a clear peak at the ground-truth: 4 clusters, again showing DECEMber capability to reveal distinct cell types if they exist.
>
> | Clusters | ARI (mean ± std) |
> |----------|------------------|
> | 2 | 0.78 ± 0.07|
> | 3 | 0.52 ± 0.31|
> | 4 | **0.95 ± 0.01**|
> | 5| 0.78 ± 0.08|
> | 6| 0.77 ± 0.15|
> | 7| 0.81 ± 0.08|
> | 8| 0.85 ± 0.05|
> | 9| 0.79 ± 0.04|
> | 10| 0.85 ± 0.07|
> | 15| 0.71 ± 0.07|
> | 20| 0.74 ± 0.06|
> | 25| 0.67 ± 0.04|
> | 30| 0.49 ± 0.02|
> | 40| 0.41 ± 0.04|
>
> That is, the marmoset retina dataset provides the positive control that our method does what it is supposed to do.
> In mouse V1 (Fig. 4), in contrast, clustering does not improve predictive performance and ARI stays below 50%, with no clear peak — suggesting a functional continuum. This negative result is meaningful and supports recent findings [1-3]. Rather than undermining the method, it highlights the method’s value for quantitatively testing, not assuming, biological structure.
>
> *(5) Methodological Issue:*
> Clipping small values is a common and practical approach to address numerical stability and it worked well in all our experiments. This is really a small detail that has no practical implication for the method.
>
> *(6) Baseline Comparison:*
> We appreciate your careful observation. You note that *“comparing an end-to-end trained method like DECEMber with a post-hoc GMM clustering on unstable embeddings is unfair”* — This is precisely the motivation for our method, because no other end-to-end method for this problem exists. Therefore, we believe that comparison with post-hoc baselines is both necessary and meaningful. If you are aware of alternative baselines that would offer a fairer comparison, we would be grateful for suggestions.
> To be clear: Across multiple DECEMber runs, the random seed of the predictive model's parameters also varies. The fact that combining training the predictive model and clustering leads to more consistent embeddings (and, hence, clustering) is one of our main contributions.
>
> *(7) Visualizations:*
> We will improve the figures and captions.
> 1 -We will add the suggested plot to the manuscript.
> As we cannot add plots for the rebuttal, below is a table. It shows that for a fixed number of clusters when β is increasing ARI increases as well and the predictive performance stays roughly constance before β>=$10^5$, where we see a rapid drop in performance without any gain in ARI.
> | ARI (mean) | Predictive performance | β | Cluster |
> |---|---|---|---|
> | 0.263 | 0.398 | 10 | 5 |
> | 0.378 | 0.396 |$10^2$| 5 |
> | 0.403 | 0.398 |$10^3$ | 5 |
> | 0.416 | 0.397 |$10^4$| 5 |
> | 0.419 | 0.395 |$10^5$| 5 |
> | 0.421 | 0.385 |$10^6$| 5 |
> | 0.422 | 0.372 |$10^7$| 5 |
> | 0.193 | 0.398 | 10 | 10 |
> | 0.267 | 0.397 |$10^2$| 10 |
> | 0.286 | 0.398 |$10^3$| 10 |
> | 0.298 | 0.396 |$10^4$| 10 |
> | 0.301 | 0.394 |$10^5$| 10 |
> | 0.304 | 0.377 |$10^6$| 10 |
> | 0.305 | 0.256 |$10^7$| 10 |
> | 0.14 | 0.398 | 10 | 20 |
> | 0.189 | 0.398 |$10^2$| 20 |
> | 0.205 | 0.397 |$10^3$| 20 |
> | 0.218 | 0.397 |$10^4$| 20 |
> | 0.221 | 0.391 |$10^5$| 20 |
> | 0.224 | 0.372 |$10^6$| 20 |
> | 0.225 | 0.352 |$10^7$| 20 |
>
> 2-4 -  Fig 7 caption will be : *“ARI across three seeds per model. A: Mouse retina [19], averaged over six models reflecting different experimental conditions (see App. A.7). B: Monkey V4 [51], three seeds. DECEMber improves ARI across both datasets and architectures, consistent with mouse V1 (Fig. 4H), supporting its generalization.”*
>
> We will extend Fig. 7A/B to 60 clusters for consistency with Fig. 4H. However, **the evaluation method is the same across Figures 4H, 7A, and 7B** - all ARI values are computed over three seeds per model. For the mouse retina, we train six separate models to account for different experimental conditions (temperature), which strongly affect response speed and would trivially separate neuron groups [3,4]. ARI curves are averaged over three seeds and weighted by neuron count. See Appendix A.7 for details and per-model ARIs.
>
> Overall, these results further support our finding: DECEMber improves consistency across diverse datasets and models, when enough neurons are available.
>
> *(8) Ambiguous Contribution:*
> We appreciate your concern as it's an important point to understand.
> First, to clarify: in Fig. 4F, the best ARI on mouse V1 is ~0.40 (not 0.25) for 5 clusters, with a baseline of ~0.25 (not 0.15) — a ~60% improvement, consistent across cluster counts.
>
> That said, we agree the absolute ARIs are low, as are other metrics (e.g., Fowlkes–Mallows score ~0.5, Appendix A.4), even when the clustering bias is strong. But this is not a model weakness — **it’s the core scientific finding**: no method, including DECEMber, reveals clear, stable clusters. This supports the idea that mouse V1 excitatory neurons form a functional continuum rather than discrete types (if discrete functional cell types existed, we’d see an ARI peak at some cluster count—not a decaying trend, as we saw in point 4 for the marmoset retina).
> We hope this reply helps to clarify our contributions,methodology, and revisit the evaluation.
>
> *References*:
> [1] Weis et al. (2025) doi: 10.1038/s41467-025-58763-w
> [2] Tong et al. (2023) doi: 10.1101/2023.11.03.565500
> [3] Weiler et al. (2023) doi: 10.1093/cercor/bhac303
> [4] Zhao et al. (2020) doi: 10.1038/s41598-020-60214-z
> [5] Höfling et al. (2024) doi: 10.7554/eLife.86860
> [6] Baden et al. (2016) doi: 10.1038/nature16468

---

> > ### Comment · Reviewer_2CDw · 2025-08-05
> >
> > Thanks for your rebuttal!
> >
> > I find that the following methods have similar aim with your work. Could you explain how you diff from them?
> >
> > 1. In vivo cell-type and brain region classification via multimodal contrastive learning ICLR 2025
> > 2. Neuron Platonic Intrinsic Representation From Dynamics Using Contrastive Learning ICLR 2025
> > 3. Learning Time-Invariant Representations for Individual Neurons from Population Dynamics NeurIPS 2024
> > 4. NetFormer: An interpretable model for recovering dynamical connectivity in neuronal population dynamics ICLR 2025

---

> > > ### Author Response · Authors · 2025-08-06
> > > **Positioning DECEMber**
> > >
> > > Thank you for engaging in the discussion!
> > >
> > > The short answer is: They all learn neuronal embeddings in some way, but none of them explicitly enforce or optimize for clustering, which is the main goal of DECEMber.
> > >
> > > Now to the details:
> > >
> > > In summary, DECEMber has two parts: a model with input- and time-invariant weight vectors for each neuron, and clustering parameters (Student-t mixture model with means and scale matrices). The regression models take images or videos as input and predict neuronal activity, tested on data from micro-electrode arrays (marmoset retina), extracellular spikes (monkey V4), and calcium imaging (mouse retina and V1). Each neuron has a weight vector (neuronal embedding), and DECEMber clusters these jointly—every neuron is compared to all others during every gradient update, unlike contrastive models that only use subsets in each batch.
> > >
> > >
> > > The works that you have mentioned differ in the following way:
> > >
> > > * As already stated at the outset, none of the works [1–4] explicitly enforce clustering.
> > > * Nemo [1] and NeurPIR [2] are contrastive methods, DECEMber is not.
> > > * Nemo [1] and NeurPIR [2] embeddings are functions of input (current activity, autocorrelogram), for DECEMber neuronal embeddings are time- and input- invariant weights of the predictive model (they embed the neuron’s full input–output function)
> > > * Nemo [1], NeuPRINT [3], NetFormer [4] do not model visual stimuli. NeuPRINT [3] and NetFormer [4] have time-invariant model weights, but both predict neuronal activity based on masked or previous neuronal activity while the regression model in our paper predicts neuronal activity based on visual stimuli. It might be interesting to integrate DECEMber clustering loss with [3] and [4] but this comes with scaling challenges (see detailed reply below).
> > >
> > > The methods in detail:
> > > * Nemo [1] is a contrastive method that generates neuronal embeddings from waveforms and autocorrelograms. Unlike DECEMber, its embeddings are not time- or input-invariant, especially when stimuli vary. Nemo does not take visual or behavioral inputs, unlike the regression models used in this work. It is designed specifically for electrophysiological signals and likely suboptimal on calcium imaging due to calcium’s low temporal resolution (~10 Hz vs 1–10 kHz). Finally, while Nemo is contrastive, DECEMber is not. Its base model performs regression from visual stimuli to neural activity, and its clustering loss doesn’t have positive-negative pairs.
> > >
> > > * NeurPIR [2] is another contrastive method, where the neuronal embeddings are functions of input. Again, DECEMber is not contrastive and neuronal embeddings are actually model weights. Unlike NEMO, NeurPIR can partially account for stimulus and behavior via CEBRA encodings.
> > >
> > > * NeuPRINT [3] is a bit closer to DECEMber compared to Nemo and NeurPIR. In this case the model has time-invariant weight embeddings but instead of predicting the neuronal activity based on visual stimuli NeuPRINT takes masked neuronal activity as input and tries to predict the masked parts. So, our base model is different as it takes visual stimuli into account while NeuPRINT does not. However, one could potentially integrate the clustering loss from DECEMber into the NeuPRINT training procedure.
> > >     * *Technical side comment:* NeuPRINT is difficult to scale to the Sensorium 2022 dataset (~50,000 neurons) because it is a transformer with a [Neurons × Neurons] attention matrix, leading to high memory use.
> > > In float16 1 digit is 2 bytes, 1Gb=$10^9$ bytes, so the memory for the forward pass is 2 * Neurons * Neurons * model dimension / $10^9$ = 5 *model dimension Gb, NeuPRINT model dimension was 32, so it is 160Gb for a single layer attention without considering more memory-intense backpropagation. For comparison, H100 has less than 100Gb.
> > >
> > > * Like NeuPRINT, NetFormer [4] uses time-invariant neuronal weight embeddings and predicts neuronal activity. While NeuPRINT masks activity, NetFormer predicts future activity from history and also infers dynamical connectivity. Unlike our regression models, NetFormer ignores visual stimuli, whereas our models predict activity from stimuli, not history. Same as for NeuPRINT, it might be interesting to integrate DECEMber clustering loss on NetFormer time-invariant embeddings but it's likely outside the rebuttal timeline as the same scaling limitations as for NeuPRINT apply for NetFormer.
> > >
> > > We hope this helps to position our method better and we will include the following works into the discussion / related works sections!
> > >
> > > References:
> > > [1] In vivo cell-type and brain region classification via multimodal contrastive learning ICLR 2025
> > > [2] Neuron Platonic Intrinsic Representation From Dynamics Using Contrastive Learning ICLR 2025
> > > [3] Learning Time-Invariant Representations for Individual Neurons from Population Dynamics NeurIPS 2024
> > > [4] NetFormer: An interpretable model for recovering dynamical connectivity in neuronal population dynamics ICLR 2025

---

### Official Review · Reviewer_ZNMh · 2025-07-02

**Clarity:** 2
**Significance:** 1
**Originality:** 1
**Rating:** 3
**Confidence:** 3

**Summary:**

This paper proposes DECEMber, a deep learning clustering model designed to learn meaningful latent neural representations. The model is trained in a two-stage process. Initially, a pre-training step establishes a feature representation using an architecture composed of a shared core and neuron-specific Gaussian readouts. This core generates a common feature map, and individual neuron responses are computed via a dot product between their weights and a selected feature vector, guided by each neuron's receptive field. Subsequently, the model is fine-tuned for clustering by introducing an additional loss to the objective function, leveraging the EM algorithm to iteratively learn and refine cluster assignments. The authors validate the clustering capabilities of DECEMber across a range of datasets, including a numerical simulation and complex biological data from the retina and visual cortex of mice and non-human primates.

**Questions:**

- The authors stated that the model avoids the trivial solutions and mode collapse issues inherent in DEC by learning a covariance matrix for each cluster. However, the mechanism by which this prevents mode collapse is not fully clear. Could the authors please elaborate on precisely how learning this covariance matrix confers this stability? For instance, if the clustering stage were initialized with a unit scale factor for all clusters, as is common, what prevents the model from converging to a trivial solution before the covariances are sufficiently learned?

- In the analysis of the marmoset retina dataset, it was noted that the model clusters neurons by their source retina, suggesting a batch effect. As a result, the authors trained each retina separately. This outcome is somewhat unexpected given the model’s architecture. A key motivation for learning a shared neural embedding in the first stage is presumably to produce representations that are invariant to nuisance factors such as batch effects. Could the authors clarify why the model, despite this design, fails to integrate data from the two retinas?

**Ethical Concerns:**

["NO or VERY MINOR ethics concerns only"]

**Final Justification:**

I will keep my score. In my opinion, the current version of the manuscript requires major revisions to more demonstrate the impact and novelty of the method, and clearly distinguish its results. At present, it is unclear how other comparable methods, such as IDEC, would perform on visual cortex datasets. The authors may need to consider adding a few well-established benchmarking datasets to showcase the superiority of their method compared to others.

**Limitations:**

The authors have already mentioned some important limitations of the proposed deep clustering method. In my view, the most critical limitations are:

•	The need to predefine the number of clusters during training. The model lacks a strategy for adaptively determining the optimal number of clusters.

•	The model's performance appears to be sensitive to the number of pretraining epochs, the $\beta$ hyperparameter, and the learning rate.

•	The proposed framework is not a sufficiently robust exploratory tool for uncovering both discrete and continuous sources of variability in the dataset.

**Paper Formatting Concerns:**

The supplementary materials appear to be incomplete or improperly uploaded.

**Quality:**

2

**Strengths And Weaknesses:**

**Strengths**

- Important Problem Domain: The paper tackles an important and challenging problem of identifying functional cell types by learning neuronal functional embeddings. This is a core question in neuroscience, and novel computational approaches are highly valuable.

- Comprehensive Datasets: The authors validate their model on a sufficient combination of numerical simulations and relevant real-world datasets.

**Weaknesses**

- Suboptimal Training Strategy: The two-stage training process that includes pre-training a core unit followed by clustering-specific fine-tuning, is a suboptimal design. This separation can prevent the model from learning features that are informative for the final clustering task from the outset. A joint, end-to-end training approach would likely yield more task-relevant embeddings. This limitation is highlighted by the ambiguity in hyperparameter tuning (e.g., Figures 5 and 7), where it is unclear how many pre-training epochs are optimal.

- Unsuited for Discovering Cluster Number:  The proposed framework is not an appropriate choice for addressing the fundamental question of how many functional cell types exist in the data or whether they are organized into discrete or continuous structures. The model itself does not provide a mechanism for model selection. The authors attempt to use the Adjusted Rand Index (ARI) for this. While the ARI measures similarity between clustering results or against a known ground truth, it does not indicate the correct number of clusters. A more appropriate method, like using the Bayesian Information Criterion (BIC), would be required to provide evidence for a specific number of clusters. I do not think the paper's claims about continuous versus discrete organization are not well-supported by its own analysis.

- Clustering Algorithm: The decision to build upon Deep Embedded Clustering (DEC) may not be the most suitable choice. Prior studies, IDEC [1] have highlighted that the clustering loss used in DEC can distort the feature space by forcing embeddings toward predefined cluster centroids. This approach risks compromising the local structure of the data, potentially resulting in learned representations that fail to preserve the intrinsic organization of the original neural activity.

- Insufficient Baselines and Performance: The comparison to other methods is lacking. The paper only includes simple baselines like GMM and K-means, while ignoring more powerful deep clustering techniques (e.g., methods based on contrastive learning like [2] or IDEC).

- The model's reported predictive performance seems low. It is primarily reported as a correlation score (Figure 4), which consistently remains below 0.4 across different experiments, raising questions about the model's practical utility.

- Supplementary Materials: There appears to have been a mistake during the upload of the supplementary material. The supplement is attached to the main manuscript file, and the separately uploaded supplementary file seems to be missing content

-------------------------
[1] Guo, Xifeng, et al. "Improved deep embedded clustering with local structure preservation." Ijcai. Vol. 17. 2017.

[2] Shen, Yuming, et al. "You never cluster alone." Advances in Neural Information Processing Systems 34 (2021): 27734-27746.

---

> ### Author Rebuttal · Authors · 2025-07-31
>
> Thank you for your detailed feedback, acknowledging that we are solving an important problem on a “comprehensive” real-world dataset.
>
> We believe that the concerns raised are mostly due to misunderstandings.
> *Let us first address the criticism (weaknesses) you raised:*
>
> - *Suboptimal Training Strategy / Clustering Algorithm:*
> **We do train jointly**. After a brief pretraining phase, the clustering loss is combined with the Poisson loss, similar in spirit to IDEC [1], thereby preserving local structure. To avoid biasing neuronal structure in any way we initialize neuronal embeddings as a constant at the beginning. Therefore, the model first needs to learn minimal feature separation for clustering to be meaningful. Additionally, IDEC also “pretrain a stacked denoising autoencoder before performing clustering”. Hence, we are already following your suggestions.
> Different optimal amounts of pretraining epochs across datasets are more likely explained by different model architectures and very different nature of data in different datasets (electrophysiology vs calcium imaging, hundreds of neurons to thousands of neurons in a model and different signal-to-noise ratios).
>
> - *Discovering Number of Clusters*:
> We would like to clarify why ARI is suitable to detect the number of clusters: ARI measures the agreement between two partitions of points by evaluating whether pairs of points are assigned to the same cluster in both partitions. When a ground-truth number of clusters exists, the ARI should peak at that value because points are consistently grouped together. This makes our clustering approach well-suited even in scenarios where the true number of clusters is not known in advance. We validated it on a marmoset retina, where N=4 is the ground truth number of clusters and we indeed see a clear ARI peak at 4 clusters.
> | Clusters | ARI (mean ± std) |
> |----------|------------------|
> | 2 | 0.78 ± 0.07|
> | 3 | 0.52 ± 0.31|
> |**4** | **0.95 ± 0.01**|
> | 5| 0.78 ± 0.08|
> | 6| 0.77 ± 0.15|
> | 7| 0.81 ± 0.08|
> | 8| 0.85 ± 0.05|
> | 9| 0.79 ± 0.04|
> | 10| 0.85 ± 0.07|
> | 15| 0.71 ± 0.07|
> | 20| 0.74 ± 0.06|
> | 25| 0.67 ± 0.04|
> | 30| 0.49 ± 0.02|
> | 40| 0.41 ± 0.04|
>
>     We will integrate this analysis in the final version of the paper.
>     As you suggested, BIC could be an alternative way to select the number of clusters.
>     As BIC requires held-out neurons to estimate the test likelihood, we did it on mouse V1 datasets (marmoset retina is too small for it as it has ~150 neurons).
>     For each mouse we randomly hold-out 200 neurons for each of the seven mice, which were trained together with the model but were ignored during the clustering procedure which fitted the TMMs. We used a model with 10 pretraining epochs and β=10^4. After the model was trained we a) computed the likelihood on the hold-out neurons and b) performed an additional E-step on the held-out neurons, calculating the soft assignments, such that we could evaluate ARI on only these 200 neurons as well.
>     We computed BIC as $BIC = -2 log L + (2 \cdot k \cdot d + 1) \cdot log(N)$, $N$ is the number of samples ($N = 200 * 7$), $d$ is the dimensionality of the embeddings ($k=128$ in our case),  $k$ is the number of clusters and $log L$ is the likelihood computed as $log L =  \log\left( \frac{1}{K} \sum_{k=1}^{K} t(x \mid \mu_k, \mathrm{var}_k, \nu) \right) $.
>     As we have not observed any significant changes in the likelihood, BIC just linearly grows with the number of clusters (number of model parameters), indicating the lack of clear cluster peak, agreeing with ARI (Fig. 4Hin the paper).
>     |Clusters|Likelihood|BIC|
>     |---|---|---|
>     |5|375.3 ± 27.8|8529 ± 55.6|
>     |10|348.5 ± 15.7|17855 ± 31.5|
>     |15|355.0 ± 34.4|27115 ± 68.7|
>     |20|335.1 ± 3.2|36427 ± 6.3|
>     |25|382.9 ± 32.2|45605 ± 64.4|
>     |30|387.2 ± 42.7|54869 ± 85.3|
>     |35|351.2 ± 30.2|64213 ± 60.5|
>     |40|371.2 ± 33.4|73446 ± 66.8|
>     |45|357.4 ± 26.6|82746 ± 53.2|
>     |50|367.7 ± 22.1|91998 ± 44.3|
>
>
> - *Insufficient Baselines:* We use post-hoc GMM/k-means clustering baselines due to the lack of more suitable alternatives. Our method already resembles IDEC as we do joint optimization to preserve local structure. We cannot use vanilla DEC since it forces unit-scale embeddings, but our embeddings are model weights whose scale  is determined by the scale of neuronal activity. Thank you for pointing us to the “You Never Cluster Alone” paper [2]. While interesting, its contrastive approach would require augmentation of model weights—our per-neuron embeddings. As it’s not clear how one would do data augmentation in this case, this would constitute a separate research project.
>
> - *Low predictive performance:* The predictive performance is not low – it’s close to the state of the art. It just reflects that predicting neuronal responses in mouse V1 is a hard task. Our numbers are comparable to the best models on Sensorium 2022 [3] – Table 3. A few percent can be gained by ensembling multiple models. We show the potential for this improvement by ensembling models across 8 seeds - this gave us 41% correlation for models with 5, 10 and 20 clusters (there were 8 seeds for each number of clusters, we did not ensemble across a number of clusters). Please note that this is a qualitative comparison as sensorium 2022 live and final test sets are not publicly available, we used test sets from 5 fully publicly available mice to compute the test metrics.
>
> - *Supplementary Materials:* We apologize for the confusion. We intended to upload all supplementary materials together with the main text to preserve hyperlink functionality, with a single plot being added later. We will integrate all materials into a single document.
>
> *With respect to your questions:*
>
> 1) *Mode collapse:* To clarify: we are not addressing general DEC mode collapse, but adapt DEC to our specific setting, where embedding scales are not arbitrary. In standard DEC, unit-scale clusters are acceptable because the latent space of the autoencoder can adapt. In our case, the embeddings to be clustered are model weights. Their scale is constrained by the scale of neuronal activity. Changing their magnitude would harm the predictive model. Therefore, we need to adjust the scale and shape of the clusters. To do so, we introduce scale matrices that let the model capture anisotropic cluster structure and non-unit scale. As noted in line 159, these matrices are initialized from empirical within-cluster variances, computed after a few pretraining epochs using only the Poisson loss—providing a data-driven, meaningful starting point.
>
> 2) *Batch effects:* As you correctly noticed, we do train several models for mouse and marmoset retinas. We do it due to the differences in the experimental conditions (presumably different temperature) which influences the speed of the responses so much that this would be the most trivial dimension to separate the neurons for different groups [4, 5]. While several works try to address this issue and integrate several retinas into the same model [6, 7, 8], this is still an active area of research, which is not the main focus of our paper.
> We also would like to note that DECEMber is not a specific model per se, but rather an additional loss that we integrate with various models. This also means that models for monkey V4, mouse V1 and mouse retina all have slightly different architecture, especially on the readout layers. And for mouse V1 and monkey V4, we do train jointly on several animals, as the experimental conditions between subjects there are comparable and DECEMber does not suffer from batch effects there.
>
>
> We will add IDEC[1] into the related works section and integrate the BIC analysis to the appendix.
>
> We hope these comments shed some light on our motivation, technical and design choices. We also hope that it would help you to reconsider some criticism.
>
> *References*:
> [1] Guo, Xifeng, et al. "Improved deep embedded clustering with local structure preservation." Ijcai. Vol. 17. 2017.
> [2] Shen, Yuming, et al. "You never cluster alone." Advances in Neural Information Processing Systems 34 (2021): 27734-27746.
> [3] Willeke, K. F., Fahey, P. G., Bashiri, M., Hansel, L., Blessing, C., Lurz, K. K., ... & Sinz, F. H. (2023, August). Retrospective on the SENSORIUM 2022 competition. In NeurIPS 2022 Competition Track (pp. 314-333). PMLR.
> [4] Zhao, Z., Klindt, D. A., Maia Chagas, A., Szatko, K. P., Rogerson, L., Protti, D. A., ... & Euler, T. (2020). The temporal structure of the inner retina at a single glance. Scientific reports, 10(1), 4399.
> [5] Fritsches, K. A., Brill, R. W., & Warrant, E. J. (2005). Warm eyes provide superior vision in swordfishes. Current Biology, 15(1), 55-58.
> [6] Höfling, L., Szatko, K. P., Behrens, C., Deng, Y., Qiu, Y., Klindt, D. A., ... & Euler, T. (2024). A chromatic feature detector in the retina signals visual context changes. Elife, 13, e86860.
> [7] Vystrčilová, M., Sridhar, S., Burg, M. F., Gollisch, T., & Ecker, A. S. (2024). Convolutional neural network models of the primate retina reveal adaptation to natural stimulus statistics. bioRxiv, 2024-03.
> [8] Gonschorek, D., Höfling, L., Szatko, K. P., Franke, K., Schubert, T., Dunn, B., ... & Euler, T. (2021). Removing inter-experimental variability from functional data in systems neuroscience. Advances in Neural Information Processing Systems, 34, 3706-3719.

---

> > ### Comment · Reviewer_ZNMh · 2025-08-03
> > **Response to authors' rebuttal**
> >
> > I thank the authors for their detailed and comprehensive response to my initial review. While the rebuttal was insightful, I want to elaborate on some key points that remain unresolved.
> >
> > **Suboptimal, sequential training strategy:** I understand that the final optimization incorporates both a model loss ($L_{model}$) and the clustering loss ($L_{cluster}$). However, as mentioned in Algorithm 1 and confirmed by the authors, the training is not performed jointly from the outset. Instead, it follows a sequential, two-stage process: an initial pre-training phase that optimizes the feature map using only the $L_{model}$, followed by clustering, which is introduced with a substantial weight.
> > My concern with this "suboptimality" is twofold:
> > - The method's reliance on a distinct pretraining phase makes the final clustering performance sensitive to the duration and quality of this initial step. This dependency suggests that the final results could vary based on the pretraining protocol, which could affect the model's robustness and reproducibility.
> > - It is unclear to what extent the learning remains "joint" after the clustering objective is introduced. With a large weight assigned to $L_{cluster}$, its gradients could dominate the optimization process. This potentially marginalize the influence of $L_{model}$, effectively stalling further refinement of the feature representation. This calls into question how much the two objectives truly influence each other in the latter stages of training.
> > This is a major concern, particularly if DECEMber is proposed as a general clustering method capable of handling diverse data modalities with varying complexity.
> >
> > **Comparison with Existing Methods**
> >
> > *Distinction from IDEC:* The authors acknowledge that their proposed method is similar to IDEC. However, IDEC is neither cited in the manuscript nor the methodological differences clearly explained in the rebuttal. If I am not mistaken, IDEC does not involve a pre-training phase for the encoder–decoder model. If the main difference is only in the application rather than in the methodology itself, it is unclear why DECEMber is needed as a distinct method.
> >
> > *Contrastive methods:* I understand that applying contrastive learning methods, which typically rely on data augmentation or a well-defined similarity metric, is not straightforward in the context of neuronal clustering. However, this is a critical point that justifies the authors' methodological choices. I recommend that the authors explicitly discuss this limitation in the manuscript to explain why other state-of-the-art clustering methods may not be viable options for their specific task.
> >
> > **BIC:** I thank the authors for providing BIC results for the V1 dataset. I agree that the Rank index is an appropriate metric for the retina dataset, where there is a well-established consensus on the true number of clusters. However, in settings where the number of clusters is unknown, or one needs to assess whether a discrete cluster structure exists at all, model selection criteria such as BIC are strongly recommended.
> >
> > One question regarding the BIC values; do they also increase monotonically for $1 \le K \le 5$? Also, when you say "$K=128$", you meant $d$, right?

---

> ### Author Response · Authors · 2025-08-04
> **Clarifying training strategy, distinction from DEC/IDEC and BIC calculations**
>
> Thanks for engaging in the discussion!
>
> **Suboptimal training**: We are happy to clarify more:
>
> - Good clustering depends on meaningful initialization without it, we risk forcing points into incorrect groups due to early guesses, leading to (random initialization) biased results. That is why **both DEC and IDEC also include pretraining** (IDEC page 3, sec. 3.1 *“Follow suggestions in [Xie et al., 2016], we also pretrain a stacked denoising autoencoder before performing clustering. After pretraining…”*).
> While we understand concerns about pretraining length affecting results, this limitation applies to all such methods. We are simply more transparent about it. Importantly, predictive performance remains stable across pretraining durations, so reproducibility is unaffected. However, internal representations are sensitive to training details (e.g., learning rate), so only models trained under the same conditions should be compared.
>
> - *“It is unclear to what extent the learning remains "joint"*:  We will clarify in the main text that for best results, β just needs to roughly balance the clustering and Poisson losses, which can be estimated with a short (1–2 epoch) run. Both losses improve significantly (clustering: $10^8 → 10^6$, DECEMber: $5×10^7 →3×10^7$, same as without clustering loss), along with both ARI and correlation improvements, indicating proper optimization for both parts.
> Whether the losses help or hinder each other is an intentional feature of DECEMber—it tests if clusters truly exist. On mouse V1, increasing β to the point where performance drops shows that forcing clusters can erase functional structure without improving consistency. **This is not a weakness of the method—it reflects the data’s continuous nature.**
> Regarding data modalities, our paper includes experiments on three distinct types—micro-electrode array (marmoset retina), extracellular spikes (monkey V4), and calcium imaging (mouse retina and V1)—which vary widely in time resolution (kHz vs Hz) and signal-to-noise ratio.
> We also show in our response to reviewer mk7Y (*signal-to-noise ratio* section) that DECEMber remains robust even in low SNR settings.
>
> **Distinction from IDEC**: In DEC and IDEC the same samples are encoded and clustered. In our case, we cluster regression weights, not input latents. The model maps images $x_i$​ to responses $y_j$ via $y_j(x_i) = \phi(x_i) * w_j$​; here, $x_i$ are regression samples, while $w_j$ (not $\phi(x_i)$ as in DEC/IDEC) are the clustering samples. This setup differs significantly from DEC/IDEC and requires substantial adaptation of DEC/IDEC—adding EM updates and a learnable scale matrix. *DEC/IDEC serve more as inspiration than direct baselines.* Moreover, in our toy example, DEC fails to separate nearby clusters, while DECEMber succeeds. And as shown in our reply to reviewer mk7Y, combining Poisson loss with DEC (e.g., IDEC) still leads to collapsed clusters, whereas DECEMber correctly separates them even in noisy settings.
>
> **Contrastive methods**: Thanks for the suggestion—we’ll include it in the discussion.
>
> **BIC**: Thanks for the BIC suggestion we will include it to the appendix.
> We are sorry, we found an error in our previous log likelihood and BIC calculations - here are the correct values on mouse V1 computed on ~53,000 neurons used in the training including models with 2-5 clusters, as you asked. Due to the large dataset, log-likelihood values are very large, causing BIC to decrease monotonously with more mixture components. Similar to ARI, BIC shows no clear peak to indicate an optimal cluster number.
>
> |Clusters|BIC ±Std|Log likelihood ±Std|
> |-|-|-|
> |2|-23482290 ±154540|11743936 ±77270|
> |3|-23622800 ±200211|11815584 ±100106|
> |4|-23897032 ±58700|11954092 ±29350|
> |5|-23972814 ±62895|11993376 ±31447|
> |10|-24391853 ±69193|12209860 ±34596|
> |15|-24349671 ±180315|12195733 ± 90157|
> |20|-24753342 ±71231|12404532 ±35615|
> |25|-24790298 ±97010|12429974 ±48505|
> |30|-24869101 ±97520|12476340 ±48760|
> |35|-24817998 ±143049|12457752 ±71524|
> |40|-24843987 ±207100|12477711 ±103550|
> |45|-24943956 ±175482|12534659 ±87741|
> |50|-25030505 ±38529|12584898 ±19264|
> > Mouse V1
>
> However, we argue that ARI is a more sensitive and suitable metric, even when the number of clusters is unknown. In our original reply in *”Discovering Number of Clusters”* section, for marmoset retina ARI shows a clear peak at 4 clusters—the known ground truth, while for BIC (table below), there is no clear peak at all, the optimal values are 2 or 3, not 4 as expected.
>
> |Cluster|BIC ±Std|Log likelihood ±Std|
> |-|-|-|
> |2|29588.08 ±387|-14155 ±194|
> |3|29585.98 ±200|-13835 ±100|
> |4|30009.33 ±171|-13728 ±86|
> |5|30588.49 ±376|-13699 ±188|
> |6|31181 ±132|-13677 ±66|
> |7|31766 ±149|-13651 ±74|
> |8|32565 ±32|-13732 ±16|
> |9|33049 ±152|-13656 ±76|
> |10|33539 ±540|-13582 ±270|
> > Marmoset retina
>
> We are sorry about the typo above, it should be *”$d$ is the dimensionality of the embeddings ($d=128$ in our case),  $k$ is the number of clusters”*.

---

### Official Review · Reviewer_mk7Y · 2025-07-03

**Clarity:** 2
**Significance:** 3
**Originality:** 2
**Rating:** 4
**Confidence:** 4

**Summary:**

The authors introduce a method for clustering neurons by their functional properties.  They call this method DECEMber - Deep Embedding Clustering via Expectation Maximization-based refinement.  This algorithm works by adding a latent t-distributed class variable onto an existing predictive model.  After pre-training on the predictive model, EM steps with the t-distributed latents are alternated with updating the model parameters.  They tested this approach with a simple toy model of linear neurons separated into two functional clusters.  They also demonstrated classification of retinal ganglion cells, where it performed extremely well.  Interestingly, application of this approach to mouse V1 data suggests no clearly definable clustering as exists in the retina.  In this case, imposing the clustering prior appears to hurt performance of the models.  They also found that the length of pretraining had an effect on the consistency of obtained clusters.

**Questions:**

Questions are all given above in Strengths and Weaknesses.

**Ethical Concerns:**

["NO or VERY MINOR ethics concerns only"]

**Final Justification:**

I stand by my original score.  This is a nice paper and an excellent contribution.  The extra analyses are nice and will add value to the paper.

**Limitations:**

The authors reasonably address limitations of their work.

**Quality:**

3

**Strengths And Weaknesses:**

The notion of cell type is a question of broad importance in neuroscience, and the ability to define cell types via functional properties (let alone connecting these with other definitions of cell type) would facilitate substantial progress.  The authors’ method is straightforward and interesting and has the advantage that it can be combined with a broad class of models.

Personally, I would have presented the model a bit differently, but this could be a matter of taste.  I found the introduction of the distributions p and q, followed by the description of the EM algorithm a bit confusing.  My approach would have been to define the latent space and modeling distributions which would be determined via the EM algorithm, and then present the update equations that are derived from that formulation via the E and M steps.  I feel like the logic of the approach would be clearer.

A potential weakness of the work is that it is unclear to what extent the underlying model can affect the results.  As an example, the neurons in the toy model are linear (and deterministic if I understand correctly?).  How does the approach perform with a reasonable model mismatch, i.e. how important is it to have an underlying model that is roughly correct?  In the case of the toy model, what would adding a stochastic term to the neuron outputs do to the results?  How would they change by the SNR of the resulting neurons?  You might easily imagine that the noisier the neurons, the more difficult the clustering problem, although on the other hand the strength of a clustering approach is the potential to share information across many neurons that have similar properties. (As the model develops the clustering, the clustering will of course feedback to the neuron models via the parameter update).  Conceptually, if you haven’t reasonably captured the appropriate set of features to describe the neurons, then clustering should be a lost cause.

To that last point:  what are the perfomances (in terms of Pearson correlation) for the toy model and the retinal models?  How do they compare to the V1 results?  Can you identify a systematic relationship between clustering and model performance? (I mean beyond the observation that imposing too strongly the clustering prior has a negative impact on performance.). For example, as mentioned above, if you decrease SNR by adding noise to the toy model, what does this do to the clustering outcomes?  Do they remain consistent?  This question is particularly important for the mouse V1 results because mouse visual neurons are extremely noisy in response to commonly used stimuli in neuroscience.

On a side note, is there a reason you treat nu (in the t-distribution) as a hyperparameter and not something to be optimized in the EM algorithm?  I have no particular objection to this but wondered if there was a technical or scientific reason.

minor:

On line 140:  “the probability of feature i”,  should this read “neuron i”?

---

> ### Author Rebuttal · Authors · 2025-07-31
>
> Thank you for your positive review, appreciating that our method addresses a *“question of broad importance in neuroscience”* and can be *“combined with a broad class of models”*.
> Below we are happy to address your questions and comments from the 3rd, 4th and 5th paragraph:
> * *Toy model from the manuscript*:
> You say, *“the neurons in the toy model are linear (and deterministic if I understand correctly?”*.
> Yes, you are correct that the toy model is linear and deterministic. Its primary purpose was not to test for robustness to noise, but to create a minimal setting where we could clearly demonstrate a specific failure mode of the original Deep Embedding Clustering (DEC) loss. The vanilla DEC loss fails because it assumes a fixed unit scale for the clusters, causing the centroids to collapse. Our toy example shows how introducing a learnable scale matrix Σj is essential to solve this problem.
> * *Signal-to-noise ratio*:
> You also suggested: *“In the case of the toy model, what would adding a stochastic term to the neuron outputs do to the results? How would they change by the SNR of the resulting neurons?”*
> As you correctly pointed out, its signal-to-noise ratio is a very important aspect of the data and as you suggested, we added a stochastic term in the toy example  to address it.
> Specifically, we generated:
>     * 1100 of images $I$, where each image is a 128-dimensional vector drawn from a standard Gaussian: $x_i \sim  \mathcal{N}(0, I_{128})$. We do a train/test split with a train set containing 90% and the test set containing 10% of the data.
>     * 1000 of neurons $N$, where the neuron weights are drawn from two Gaussian distributions to form uneven clusters: first “cell type” as  $w_j \sim$ $ \mathcal{N}($**0**$, I_{128}),  \text{for } j = 1, \dots, 700$, and second “cell type” as $w_j \sim$ $ \mathcal{N}($**1**$, I_{128}),  \text{for } j = 1, \dots, j = 701, \dots, 1000$
> The clean neural responses were simulated as a dot product between the images $I$ and neurons $N$.
> To simulate noisy observations, we added Gaussian noise independently for each neuron and image: $\boldsymbol{\varepsilon} \sim \mathcal{N}(0, \sigma^2)$.
> The noisy responses are thus given by: $y_{ij} = w_j^T x_i + \varepsilon_{ij}$.
> Since both the input $\mathbf{I}$ and the noise $\boldsymbol{\varepsilon}$ are mean-centered, the Signal-to-Noise Ratio of neuron $j$ (SNR_j) is defined as:
> $\mathrm{SNR}_j =$ $\frac{\operatorname{Var}_i(w_j^T x_i)}{\operatorname{Var}_i(\varepsilon_ij)}$ (both $i$ and $j$ are indexes for $\varepsilon$ but openreview struggles to display it correctly).
> In this setup, we vary the SNR by adjusting the noise variance $\sigma^2$, thereby controlling the noise power in the simulation.
> We varied the SNR between 0.001 and 10 and same as in the paper toy example we first pretrained the regression model on MSE for 30 epochs and then finetuned with DEC or DECEMber + MSE loss for 150 more epochs. We calculated the Pearson correlation on a left out test set. Both DEC and DECEMber converge to similar performance since the MSE drives the learning of the model’s weights.
> Below you can find the table with the distance between cluster centers given different SNRS (the ground truth distance between the cluster centers is $\sqrt{128}  \sim 11.3$).
> **We see that even for low SNR DECEMber successfully separates the clusters**, though the distance is not ideal before SNR is above 1.
> However, for DEC cluster collapse happens independently of SNR.
> Since the weights (e.g. neurons) are generated based on predefined clusters (means of the Gaussians), we know the ground truth label for each neuron. To evaluate the performance of DEC and DECEMber we measure the proportion of neurons that are assigned to the correct cluster.
> |SNR|Distance between cluster centers||Predictive accuracy (correlation)||Cell classification accuracy||
> |---|---|---|---|---|---|---|
> ||DEC|DECEMBER|DEC|DECEMBER|DEC|DECEMBER|
> |0.001|1.968|3.908|0.000|0.000|51.6|50.1|
> |0.01|1.213|4.370|0.022|0.025|61.0|79.0|
> |0.05|0.197|6.993|0.100|0.109|79.0|98.1|
> |0.1|0.048|8.828|0.180|0.189|59.2|99.8|
> |0.5|0.015|10.494|0.489|0.494|51.4|100|
> |1|0.007|10.76|0.641|0.642|52.6|100|
> |2|0.011|10.95|0.771|0.771|50.2|100|
> |5|0.010|11.058|0.890|0.890|51.5|100|
> |10|0.015|11.075|0.941|0.940|50.3|100|
>
> * *Relationship Between Model Performance and Clustering:*
> You asked *“Can you identify a systematic relationship between clustering and model performance?”*, which is a very interesting point.
> You are right, *“conceptually, if we haven’t reasonably captured the appropriate set of features to describe the neurons, then clustering should be a lost cause”*.
> E.g. on a toy example for the smallest SNR=0.001 the cell types accuracy is worse than a random guess (the probability that a random guess matches the true label
> $P(\text{predict}=i) \times P(\text{true label}=i) = 0.7^2 + 0.3^2 = 0.58$).
> However, even the minimal catch of meaningful signal already significantly improves the clustering accuracy for DECEMber (SNR=0.01 gives 2.5% correlation and 79% cell types classification accuracy). And of course, better model performance helps to improve the clustering: On the toy example above we see the cell types classification accuracy improving up to SNR=0.5, where it saturates as it reaches the best possible score.
> **The models we use for real data, including mouse V1, are close to the state-of-the-art** [2] and actively used for closed-loop [3] or in-silico experiments [4,5]. Therefore, we believe they do capture meaningful features even though the correlations are far from 1.
> The fact that DECEMber improves cluster consistency on the very noisy mouse V1 dataset provides strong evidence that DECEMber is effective in the high-noise, low-SNR regime that is typical for in-vivo neuroscience data, even though the models are not perfect.
> You also state that *“the strength of a clustering approach is the potential to share information across many neurons that have similar properties.”* That’s correct only if the clusters are very homogeneous. If there is a lot of within-cluster variance, there is not so much information to share. Pulling all neurons towards the cluster mean would actually hurt the performance, because the neurons indeed are different.
> You also mentioned in your summary that *“imposing the clustering prior appears to hurt performance of the models.”* It doesn’t actually hurt performance unless we overdo it. For smaller beta~=10^4 (Fig. 4H), predictive performance is maintained but embedding consistency improves substantially. The experiment of increasing the clustering weight, β, to the point where performance drops is intentional. It demonstrates that aggressively forcing a clustered structure can erase functionally relevant details, but does not improve embeddings’ consistency. This result supports our conclusion that excitatory neurons in the mouse visual cortex likely form a continuum rather than discrete types. **It is not a weakness of the method – it is a property of the data!**
>
> * *$\nu$ as a Hyperparameter:*
> In the 5th paragraph, you ask if there is a reason we *”treat* $\nu$ *(in the t-distribution) as a hyperparameter”*.
> While it is possible to optimize ν in a t-mixture model, it does not have a simple closed-form update in the M-step, unlike the mean and scale matrices. We tried both having as a learnt hyperparameter on toy data and also explored the values between 2-20 on mouse V1 data, however, it had little to no impact on clustering outcomes or overall model performance, which is consistent with the original DEC paper [1], so we chose to fix $\nu$ as a hyperparameter for simplicity and computational efficiency. We will add a section in the appendix summarizing these findings.
> |ARI ± std|clusters|$\nu$|
> |---|---|---|
> |0.416 ± 0.012|5|2.1|
> |0.417 ± 0.013||5|
> |0.418 ± 0.013||10|
> |0.42 ± 0.012||20|
> |0.299 ± 0.025|10|2.1|
> |0.299 ± 0.025||5|
> |0.301 ± 0.025||10|
> |0.303 ± 0.027||20|
>
> * *Line 140*: yes, “feature i” is equivalent to “neuron i”
>
> We thank you for the biologically meaningful questions and helping us to improve our toy model. We will integrate it into the final manuscript.
> We are happy to address any follow-up questions as well.
>
> *References*:
> [1] Xie, J., Girshick, R., & Farhadi, A. (2016, June). Unsupervised deep embedding for clustering analysis. In International conference on machine learning (pp. 478-487). PMLR.
> [2] Willeke, K. F., Fahey, P. G., Bashiri, M., Hansel, L., Blessing, C., Lurz, K. K., ... & Sinz, F. H. (2023, August). Retrospective on the SENSORIUM 2022 competition. In NeurIPS 2022 Competition Track (pp. 314-333). PMLR.
> [3] Walker, E. Y., Sinz, F. H., Cobos, E., Muhammad, T., Froudarakis, E., Fahey, P. G., ... & Tolias, A. S. (2019). Inception loops discover what excites neurons most using deep predictive models. Nature neuroscience, 22(12), 2060-2065.
> [4] Ustyuzhaninov, I., Burg, M. F., Cadena, S. A., Fu, J., Muhammad, T., Ponder, K., ... & Ecker, A. S. (2022). Digital twin reveals combinatorial code of non-linear computations in the mouse primary visual cortex. BioRxiv, 2022-02.

---

> > ### Comment · Reviewer_mk7Y · 2025-08-04
> >
> > Thank you for the excellent and detailed reply.

---

### Decision · Program_Chairs · 2025-09-17

**Decision:**

Accept (poster)

**Comment:**

This paper proposes a deep learning technique to train a predictive model of visual neurons of the brain while simultaneously optimizing for cell type clustering. The authors demonstrate that their model is effective in both predicting neural responses and clustering cell types, with evidence showing generalization across species and brain areas.

The reviewers provided thorough evaluations that identified both strengths and limitations of this work. While there are certainly areas for improvement (additional baseline models, better visualizations, etc.) the proposed method demonstrates clear value and effectiveness despite its straightforward approach. The dual optimization for neural prediction and cell type clustering is a meaningful contribution that addresses an important challenge in neuroscience.

Some reviewers expressed concerns about methodological limitations and presentation quality. However, after careful consideration of both the reviewers' comments and the authors' rebuttals, the core contributions appear sound and the experimental validation adequate.

Given the significance of this problem and the demonstrated effectiveness of the proposed approach, I believe this work would provide value to neuroscience. The method offers a practical tool that researchers can incorporate into their analysis pipelines and test rigorously in diverse applications. While the work may represent an incremental advance rather than a major breakthrough, such contributions are essential for building the methodological foundation needed for progress in computational neuroscience.

Therefore, I recommend acceptance.